# Multimodal neural correlates of childhood psychopathology

**Jessica Royer[1†], Valeria Kebets[1,2,3,4†], Camille Piguet[5,6], Jianzhong Chen[2,3,4], Leon Qi Rong Ooi[2,3,4], Matthias Kirschner[1,7], Vanessa Siffredi[8,9,10], Bratislav Misic[1], BT Thomas Yeo[2,3,4,11,12*†], Boris C Bernhardt[1*]**

[1]McConnell Brain Imaging Centre, Montreal Neurological Institute, McGill University, Montreal, Canada; [2]Centre for Sleep and Cognition & Centre for Translational MR Research, Yong Loo Lin School of Medicine, National University of Singapore, Singapore, Singapore; [3]Department of Electrical and Computer Engineering, National University of Singapore, Singapore, Singapore; [4]N.1 Institute for Health & Institute for Digital Medicine, National University of Singapore, Singapore, Singapore; [5]Young Adult Unit, Psychiatric Specialities Division, Geneva University Hospitals and Department of Psychiatry, Faculty of Medicine, University of Geneva, Geneva, Switzerland; [6]Adolescent Unit, Division of General Paediatric, Department of Paediatrics, Gynaecology and Obstetrics, Geneva University Hospitals, Geneva, Switzerland; [7]Division of Adult Psychiatry, Department of Psychiatry, Geneva University Hospitals, Geneva, Switzerland; [8]Division of Development and Growth, Department of Paediatrics, Gynaecology and Obstetrics, Geneva University Hospitals and University of Geneva, Geneva, Switzerland; [9]Neuro-X Institute, Ecole Polytechnique Fédérale de Lausanne, Geneva, Switzerland; [10]Department of Radiology and Medical Informatics, Faculty of Medicine, University of Geneva, Geneva, Switzerland; [11]Integrative Sciences and Engineering Programme, National University Singapore, Singapore, Singapore; [12]Martinos Center for Biomedical Imaging, Massachusetts General Hospital, Boston, United States

**\*For correspondence:**
thomas.yeo@nus.edu.sg (BTTY);
boris.bernhardt@mcgill.ca (BCB)

†These authors contributed equally to this work

**Competing interest:** The authors declare that no competing interests exist.

**Abstract** Complex structural and functional changes occurring in typical and atypical development necessitate multidimensional approaches to better understand the risk of developing psychopathology. Here, we simultaneously examined structural and functional brain network patterns in relation to dimensions of psychopathology in the Adolescent Brain Cognitive Development (ABCD) dataset. Several components were identified, recapitulating the psychopathology hierarchy, with the general psychopathology (*p*) factor explaining most covariance with multimodal imaging features, while the internalizing, externalizing, and neurodevelopmental dimensions were each associated with distinct morphological and functional connectivity signatures. Connectivity signatures associated with the *p* factor and neurodevelopmental dimensions followed the sensory-to-transmodal axis of cortical organization, which is related to the emergence of complex cognition and risk for psychopathology. Results were consistent in two separate data subsamples and robust to variations in analytical parameters. Although model parameters yielded statistically significant brain–behavior associations in unseen data, generalizability of the model was rather limited for all three latent components (*r* change from within- to out-of-sample statistics: $LC1_{within} = 0.36$, $LC1_{out} = 0.03$; $LC2_{within} = 0.34$, $LC2_{out} = 0.05$; $LC3_{within} = 0.35$, $LC3_{out} = 0.07$). Our findings help in better understanding biological mechanisms underpinning dimensions of psychopathology, and could provide brain-based vulnerability markers.

## Editor's evaluation

This important study provides evidence for associations between transdiagnostic psychiatric symptom domains and brain structure and function in a large cohort. The evidence supporting the findings is solid in that brain-behaviour associations are validated in separate subsamples of the data, although out-of-sample accuracies are modest. This study will be of broad interest to researchers interested in the neurobiological basis of mental disorders.

## Introduction

Late childhood is a period of major neurodevelopmental changes (*Goddings et al., 2014*; *Lebel and Beaulieu, 2011*; *Mills et al., 2021*; *Paus et al., 2008*; *Raznahan et al., 2011*), which makes it particularly vulnerable for the emergence of mental illness. Indeed, about 35% of mental illnesses begin prior to age 14 (*Solmi et al., 2022*), motivating efforts to identify vulnerability markers of psychopathology early on *Lynch et al., 2021*. This is complemented by ongoing efforts in moving toward a neurobiologically based characterization of psychopathology. One key initiative is the Research Domain Criteria (RDoC), a transdiagnostic framework to study the neurobiological underpinnings of dimensional constructs, by integrating findings from genetics, cognitive neuroscience, and neuroimaging (*Cuthbert, 2014*; *Insel et al., 2010*). From a neurodevelopmental perspective, a transdiagnostic approach in characterizing behavioral difficulties in children and adolescents might capture a broader subset of children at risk (*Astle et al., 2022*; *Casey et al., 2014*; *Jones and Astle, 2021*; *Siugzdaite et al., 2020*). Such approach is also in line with continuum models, which have gained momentum in the conceptualization of psychiatric and neurodevelopmental conditions in recent years. While not without controversy, several neurodevelopmental conditions have been increasingly conceptualized as a continuum that encompasses subclinical expressions within the general population, intermediate outcomes, and a full diagnosis at the severe tail of the distribution (*Abu-Akel et al., 2019*; *Lundström et al., 2012*; *Robinson et al., 2016*; *Robinson et al., 2011*). Such a more quantitative approach to psychopathology could capture the entire range of variation (i.e., typical, subclinical, and atypical) in both symptom and brain data (*Insel et al., 2010*; *Plomin et al., 2009*), and help elucidate their ties (*Parkes et al., 2020*).

Psychopathology can be conceptualized along a hierarchical structure, with a general psychopathology (or *p* factor) at the apex, reflecting an individual's susceptibility to develop any common form of psychopathology (*Caspi et al., 2014*; *Kotov et al., 2017*; *Lahey et al., 2017*). Next in this hierarchy are higher-order dimensions underpinning internalizing behaviors, such as anxiety or depressive symptoms, as well as externalizing behaviors, characterized by rule-breaking and aggressive behavior. Furthermore, a neurodevelopmental dimension has been described to encompass symptoms with shared genetic vulnerability, such as attention deficit/hyperactivity deficit (ADHD)-related symptoms (e.g., inattention and hyperactivity), as well as clumsiness and autistic-like traits. This dimension is particularly relevant as it might underpin the normal variation in ADHD- and autism spectrum disorder-like traits in the general population, but also learning disabilities (*Holmes et al., 2021*). Recently, five dimensions of psychopathology, that is, internalizing, externalizing, neurodevelopmental, detachment, and somatoform (*Michelini et al., 2019*), were derived using exploratory factor analysis on parent-reported behavioral data from the Adolescent Brain Cognitive Development (ABCD) dataset, a large community-based cohort of typically developing children (*Casey et al., 2018*). These findings are largely consistent with the Hierarchical Taxonomy of Psychopathology (HiTOP) (*Kotov et al., 2017*), a dimensional classification system that aims to provide more robust clinical targets than traditional taxonomies.

Progress of neuroimaging techniques, particularly magnetic resonance imaging (MRI), has enabled the investigation of pathological mechanisms in vivo. To date, most neuroimaging studies have employed case–control comparisons between cohorts with a psychiatric diagnosis and neurotypical controls (*Etkin, 2019*). However, an increasing number of studies have adopted transdiagnostic neuroimaging designs in recent years (*Baker et al., 2019*; *Elliott et al., 2018*; *Kebets et al., 2021*; *Kebets et al., 2019*; *Parkes et al., 2021*; *Romer and Pizzagalli, 2022*; *Xia et al., 2018*). At the level of neuroimaging measures, many studies have focused on structural metrics, such as cortical thickness, volume, surface area, or diffusion MRI derived measures of fiber architecture (*Cauda et al., 2018*; *de Lange et al., 2019*; *Goodkind et al., 2015*; *Hettwer et al., 2022*). On the other hand, there has

been a rise in studies assessing functional substrates, notably work based on resting-state functional connectivity (RSFC) (*Karcher et al., 2021*; *Sha et al., 2018*; *Xia et al., 2018*). Despite increasing availability of multimodal datasets (*Casey et al., 2018*; *Jernigan et al., 2016*; *Miller et al., 2016*; *Royer et al., 2022*; *Satterthwaite et al., 2016*; *Thompson et al., 2014*; *Van Essen et al., 2013*), combined assessments of structural and functional substrates of psychopathology remain scarce, specifically with a transdiagnostic design.

In this context, unsupervised techniques such as partial least squares (PLS) or canonical correlation analysis, may provide a data-driven integration of different imaging measures, and allow for the identification of dimensional substrates of psychopathology along with potential neurobiological underpinnings. Recent work integrating neuroanatomical, neurodevelopmental, and psychiatric data has furthermore pointed to a particular importance of the progressive differentiation between sensory/motor systems and transmodal association cortices, also referred to as sensory-to-transmodal or sensorimotor-to-association axis of cortical organization (*Huntenburg et al., 2018*; *Margulies et al., 2016*; *Paquola et al., 2019*; *Park et al., 2022b*; *Sydnor et al., 2021*). Indeed, compared to sensory and motor regions, transmodal association systems, such as the default mode network, have a long maturation time, which renders them particularly vulnerable for development of psychopathology (*Paquola et al., 2019*; *Park et al., 2022b*; *Sydnor et al., 2021*). Crucially, the maturation of association cortices underlies important changes in cognition, affect, and behavior, and has been suggested to highly contribute to inter-individual differences in functioning and risk for psychiatric disorders (*Sydnor et al., 2021*).

Here, we simultaneously delineated structural and functional brain patterns related to dimensions of psychopathology in a large cohort of children aged 9–11 years old. Child psychopathology was characterized with the parent-reported Child Behavior Checklist (CBCL) (*Achenbach and Rescorla, 2013*). We favored the item-based version to capture the covariation between symptoms with more granularity compared to subscales. To profile neural substrates, we combined multiple intrinsic measures of brain structure (i.e., cortical surface area, thickness, and volume) and functional connectivity at rest in our primary analysis. Post hoc analyses in smaller subsamples also incorporated diffusion-based measures of fiber architecture (i.e., fractional anisotropy [FA] and mean diffusivity [MD]) and explored connectivity during tasks tapping into executive and reward processes. We expected that the synergistic incorporation of multiple brain measures may capture multiple scales of brain organization during this critical developmental moment, and offer sensitivity in identifying neural signatures of psychopathology dimensions. We further examined if substrates of psychopathology followed the sensory-to-transmodal axis of cortical organization. We conducted our analysis in a Discovery subsample of the ABCD cohort, and validated all findings in a Replication subsample from the same cohort. Multiple sensitivity and robustness analyses verified consistency of our findings. Applying model statistics derived from the Discovery cohort to unseen data of the Replication cohort revealed, however, low generalizability and explained variance in brain–behavior relationships. As such, we report both within-sample and out-of-sample statistics in describing patterns of findings of each latent component.

## Results

### Overview of analysis workflow

We divided a fully preprocessed and quality controlled subsample of the ABCD dataset that had structural and resting-state functional MRI (fMRI) data available into Discovery (N = 3504, i.e., 2/3 of the dataset) and Replication (N = 1747, i.e., 1/3 of the dataset) subsamples, using randomized data partitioning with both subsamples being matched on age, sex, ethnicity, acquisition site, and overall psychopathology (i.e., scores of the first principal component derived from the 118 items of the CBCL). After applying dimensionality reduction to imaging features, we ran a PLS analysis in the Discovery subsample to associate imaging phenotypes and CBCL items (*Figure 1a–c*). Significant components, identified using permutation testing, were comprehensively described, and we assessed associations to sensory-to-transmodal functional gradient organization for macroscale contextualization. We furthermore related findings to initially held out measures of white matter architecture and task-based fMRI patterns that were available in subsets of participants. Finally, we repeated our analyses in the Replication subsample and assessed generalizability when using Discovery-derived loadings in the

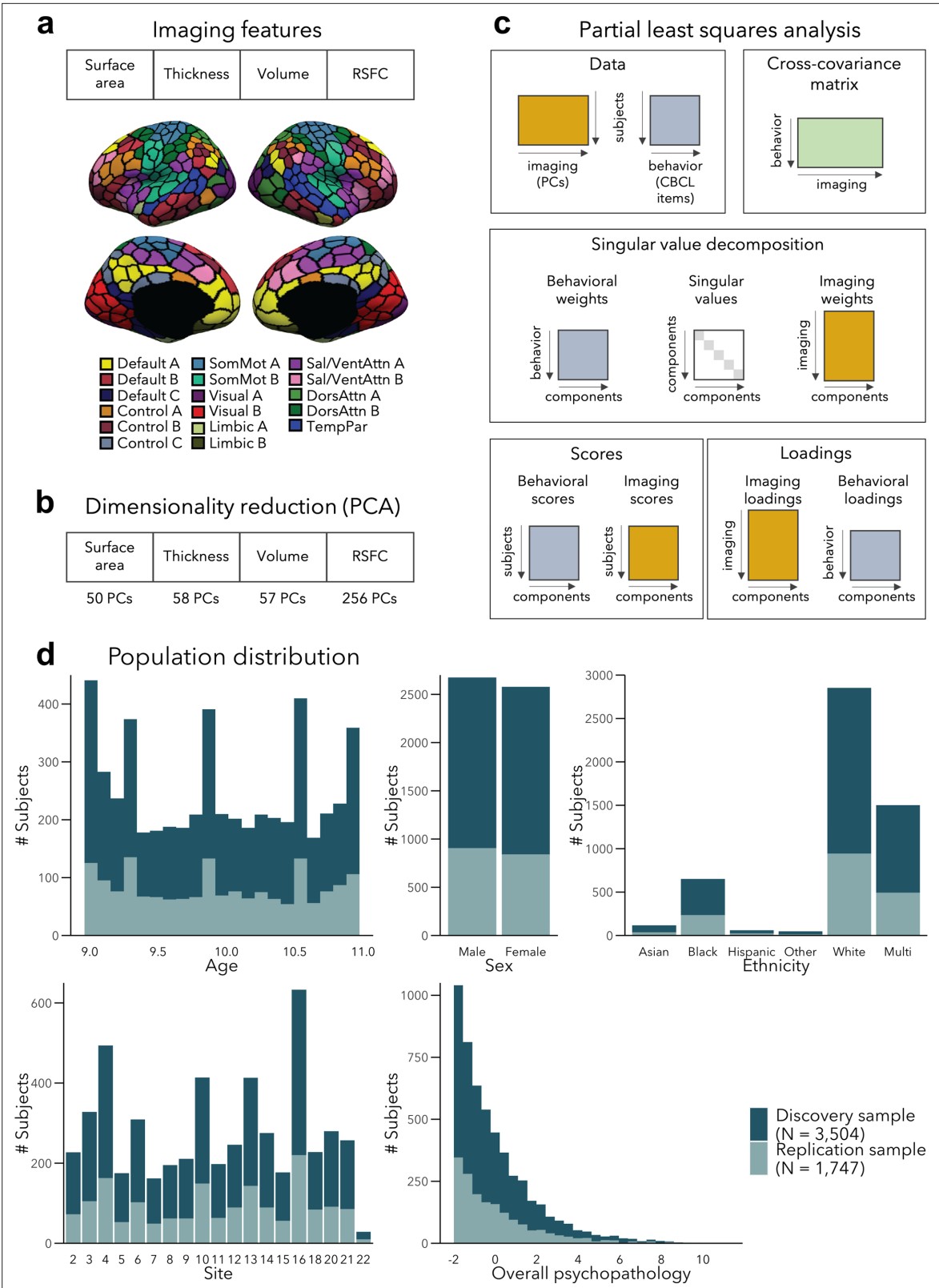

**Figure 1.** Analysis workflow. (**a**) Imaging features. (**b**) The imaging data underwent dimensionality reduction using principal components analysis (PCA), keeping the components explaining 50% of the variance within each imaging modality, resulting in 421 components in total. (**c**) Partial least squares analysis between the multimodal imaging data (421 principal components [PCs]) and the behavioral data (118 Child Behavior Checklist [CBCL] items). (**d**)

*Figure 1 continued on next page*

*Figure 1 continued*

Distribution of age, sex, ethnicity, acquisition site, and overall psychopathology were matched between the discovery and replication samples. Overall psychopathology represents the first principal component derived from all the CBCL items used in the main analysis.

The online version of this article includes the following figure supplement(s) for figure 1:

**Figure supplement 1.** Explained covariance by LCs 1–5.

**Figure supplement 2.** Post hoc analyses testing for sex differences in the composite scores between male and female participants.

**Figure supplement 3.** Relative importance of imaging modalities in LCs 1–3.

Replication subsample. Measures of brain structure and resting-state fMRI were chosen for the main analyses as they (1) have been acquired in the majority of ABCD subjects, (2) represent some of the most frequently acquired, and widely studied imaging phenotypes, and (3) profile intrinsic gray matter network organization. Nevertheless, we also conducted post hoc analyses in smaller subsamples based on diffusion-based measures of fiber architecture (i.e., FA and MD) and functional connectivity during tasks tapping into executive and reward processes. Of note, the findings presented in this paper focus on within-sample statistics, although out-of-sample statistics are reported for LC1–3 in their respective subsections. We found low generalizability of model statistics to unseen data, which was likely due to sample-specific variations in structural and functional imaging features, indicating a small amount of explained variance in brain–behavior relationships.

## Significant latent components

PLS analysis revealed five significant LCs (all p = 0.001 after permuting the first five LCs 10,000 times, accounting for site and false discovery rate [FDR]) in the discovery sample. They explained 21%, 4%, 3%, 3%, and 2% of covariance between the imaging and behavioral data, respectively (***Figure 1—figure supplement 1***). LC1, LC2, LC3, and LC5 recapitulated the dimensions previously reported (***Michelini et al., 2019***), that is, general psychopathology (LC1), internalizing vs. externalizing (LC2), neurodevelopmental (LC3), and detachment (LC5) (***Supplementary file 1a***). For the remainder of this article, we focus on LC1–LC3, as they were strongly correlated to previously identified factors (see ***Supplementary file 1a*** for details), and have been more thoroughly documented previously (***Holmes et al., 2021***; ***Michelini et al., 2019***; ***Modabbernia et al., 2022***). LC1–LC3 remained statistically significant when using within- and out-of-sample statistics, although out-of-sample generalizability of was low overall (range of $r$ = 0.03–0.07 for the first three latent components).

## General psychopathology component (LC1)

LC1 ($r$ = 0.36, permuted p < 0.001; out-of-sample generalizability of model statistics: $r$ = 0.03, permuted p < 0.001; ***Figure 2a***) recapitulated a previously described $p$ factor (***Alnæs et al., 2018***; ***Caspi et al., 2014***; ***Elliott et al., 2018***; ***Kaczkurkin et al., 2018***; ***Kebets et al., 2019***; ***Lahey et al., 2011***; ***Romer et al., 2018***; ***Shanmugan et al., 2016***; ***Van Dam et al., 2017***), and strongly correlated ($r$ = 0.64, p < 0.001; ***Supplementary file 1a***) with the $p$ factor derived from CBCL items by ***Michelini et al., 2019***. All symptom items loaded positively on LC1 – which is expected given prior data showing that every prevalent mental disorder loads positively on the $p$ factor (***Lahey et al., 2012***). The top behavioral loadings included being inattentive/distracted, impulsive behavior, mood changes, rule breaking, and arguing (***Figure 2b***, see ***Supplementary file 1b*** for all behavior loadings). Greater (i.e., worse) psychopathology was, overall, mainly associated with volume and thickness reductions, while the pattern of surface area associations was more mixed encompassing increases as well as decreases (***Figure 2c***, see ***Figure 2—figure supplement 1*** for uncorrected structural imaging loadings, and ***Figure 2—figure supplement 2*** for subcortical volume loadings). Greater psychopathology was also associated with patterns of large-scale network organization (***Schaefer et al., 2018***; ***Yeo et al., 2011***), namely increased RSFC between the default and executive control, default, dorsal and ventral attention networks, and decreased RSFC between the two attention networks, between visual and default networks, and between control and attention networks (***Figure 2d***, see also ***Figure 2—figure supplement 2*** for a zoom on subcortical–cortical loadings, and ***Figure 2—figure supplement 3*** for uncorrected RSFC loadings). Spatial correlations between modality-specific loadings are reported in ***Supplementary file 1c***.

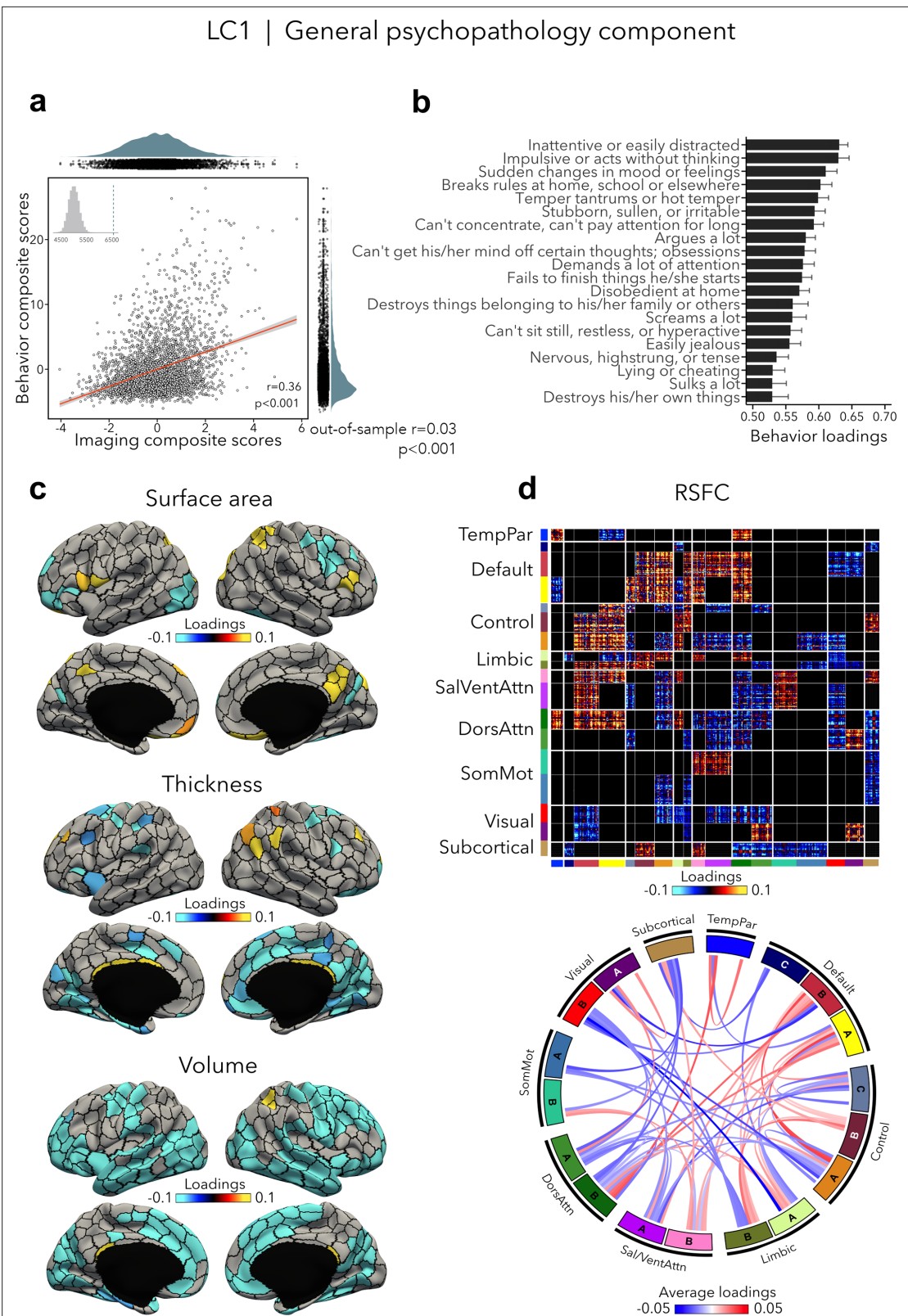

**Figure 2.** LC1 represents the general psychopathology (*p*) factor. (**a**) Correlation between imaging and behavior composite scores (*r* = 0.36, permuted p < 0.001; out-of-sample generalizability of model statistics: *r* = 0.03, permuted p < 0.001). Each dot represents a different participant from the discovery sample (n=3,504). The inset on the top left shows the null distribution of (permuted) singular values from the permutation test, while the dotted line shows the original singular value. (**b**) Top behavior loadings characterizing this component. Higher scores represent higher (i.e., worse) symptom

*Figure 2 continued on next page*

*Figure 2 continued*

severity. Error bars indicate bootstrap-estimated confidence intervals. (**c**) Significant surface area, thickness, and volume loadings (after bootstrap resampling and FDR correction *q* < 0.05) associated with LC1. (**d**) Significant RSFC loadings (after bootstrap resampling and FDR correction *q* < 0.05) associated with LC1. RSFC loadings were thresholded, whereby only within- or between-network blocks with significant bootstrapped *Z*-scores are shown. Network blocks follow the colors associated with the 17 Yeo networks (*Schaefer et al., 2018*; *Yeo et al., 2011*) and subcortical regions (*Fischl et al., 2002*). Chord diagram summarizing significant within- and between-network RSFC loadings. See also *Figure 1a* for more detailed network visualization. DorsAttn, dorsal attention; RSFC, resting-state functional connectivity; SalVentAttn, salience/ventral attention; SomMot, somatosensory-motor; TempPar, temporoparietal.

The online version of this article includes the following figure supplement(s) for figure 2:

**Figure supplement 1.** Structural loadings associated with LCs 1–3, before FDR correction.

**Figure supplement 2.** Subcortical volume loadings, and subcortical–cortical functional connectivity (FC) loadings during resting-state and the three functional magnetic resonance imaging (fMRI) tasks (Monetary Incentive Delay, Emotional n-back, and Stop Signal Task).

**Figure supplement 3.** FC patterns during rest and three functional magnetic resonance imaging (fMRI) tasks associated with LCs 1–3, before FDR correction.

## Internalizing vs. externalizing component (LC2)

LC2 (*r* = 0.34, permuted p < 0.001; out-of-sample generalizability of model statistics: *r* = 0.05, permuted p < 0.001; *Figure 3a*) contrasted internalizing vs. externalizing symptoms – two broad dimensions that are driven by covariation of symptoms among internalizing and externalizing disorders (*Lahey et al., 2012*; *Lahey et al., 2011*). Here, higher (i.e., positive) behavior loadings indicated increased internalizing symptoms, such as fear or anxiety, worrying, and feeling self-conscious, while lower (i.e., negative) behavior loadings expressed increased externalizing symptoms, such as aggressivity and rule-breaking behaviors (*Figure 3b*, *Supplementary file 1b*). At the brain level, greater internalizing symptoms were associated with widespread decreases in cortical thickness, and mixed patterns of increases and decreases in surface area and volume (*Figure 3c*). In terms of brain organization, worse internalizing symptoms reflected lower RSFC within the somatomotor network and between the somatomotor and attention networks, and higher RSFC between the somatomotor network and the default and control networks (*Figure 3d*). Spatial correlations between modality-specific loadings are reported in *Supplementary file 1c*. Note that these brain patterns were inversely related to greater/worse externalizing symptoms, for example, increased thickness, higher RSFC within the somato-motor network.

## Neurodevelopmental component (LC3)

LC3 (*r* = 0.35, permuted p < 0.001; out-of-sample generalizability of model statistics: *r* = 0.07, permuted p < 0.001; *Figure 4a*) was driven by neurodevelopmental symptoms, such as concentration difficulties and inattention, daydreaming, and restlessness (*Figure 4b*, *Supplementary file 1b*), which were contrasted to a mix of symptoms characterized by emotion dysregulation. Greater neurodevelopmental symptoms were associated with increased surface area and volume in temporoparietal regions and decreased surface area and volume in prefrontal as well as in occipital regions, but also with increased thickness in temporo-occipital areas and decreased thickness in prefrontal regions (*Figure 4c*). Worse neurodevelopmental symptomatology was also related to decreased RSFC within most cortical networks such as the default, control, dorsal, and ventral attention, and somatomotor networks, and increased RSFC between control, default, and limbic networks on the one side, and attention and somatomotor networks on the other side (*Figure 4d*). Spatial correlations between modality-specific loadings are reported in *Supplementary file 1c*.

## Association with functional gradient

The primary functional gradient explained 28% of the RSFC variance, and differentiated primary somatosensory/motor and visual areas from transmodal association cortices (*Figure 5b*). These results replicated previous findings obtained with RSFC in a healthy adult cohort (*Margulies et al., 2016*). Since previous research had found that the sensory-to-transmodal gradient only becomes the 'principal' gradient around 12–13 years old (*Dong et al., 2021*), we also computed gradients without aligning it to the Human Connectome Project (HCP) gradient to verify whether the gradient order would change, but found the principal gradient to be virtually identical to its aligned counterpart

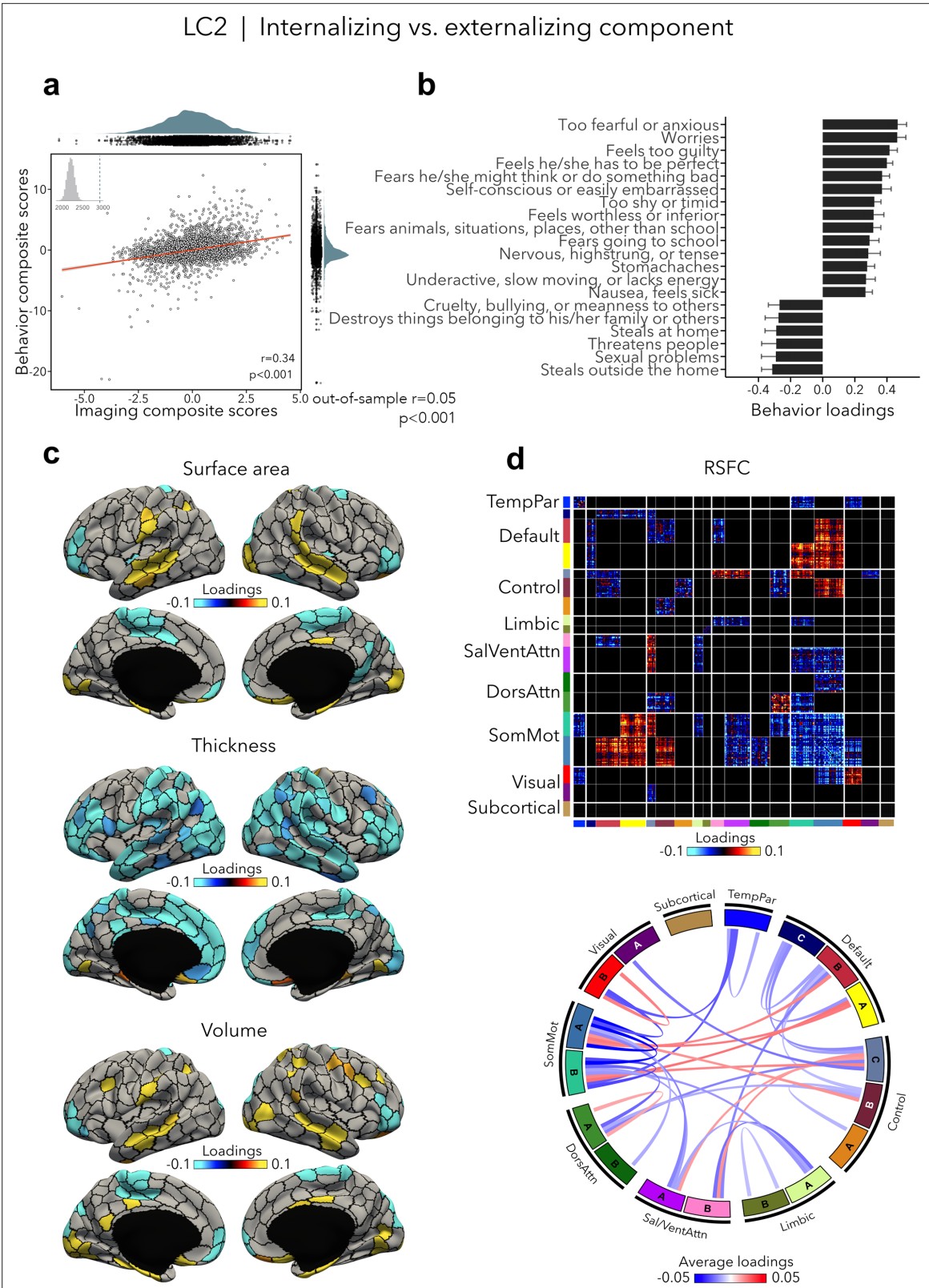

**Figure 3.** Internalizing vs. externalizing component (LC2). (**a**) Correlation between imaging and behavior composite scores (*r* = 0.34, permuted p < 0.001; out-of-sample generalizability of model statistics: *r* = 0.05, permuted p < 0.001). Each dot represents a different participant from the discovery sample (n=3,504). The inset on the top left shows the null distribution of (permuted) singular values from the permutation test, while the dotted line shows the original singular value. (**b**) Top absolute behavior loadings characterizing this component. Higher (positive) loadings represent increased

*Figure 3 continued on next page*

*Figure 3 continued*

(i.e., worse) internalizing symptoms, while lower (negative) loadings represent worse externalizing symptoms. Error bars indicate bootstrap-estimated confidence intervals. (**c**) Significant surface area, thickness, and volume loadings (after bootstrap resampling and FDR correction $q < 0.05$) associated with LC2. (**d**) Significant RSFC loadings were thresholded, whereby only within- or between-network blocks with significant bootstrapped $Z$-scores are shown. Network blocks following the colors associated with the 17 Yeo networks (**Schaefer et al., 2018**; **Yeo et al., 2011**) and subcortical regions (**Fischl et al., 2002**). Chord diagram summarizing significant within- and between-network RSFC loadings. See also **Figure 1a** for more detailed network visualization. DorsAttn, dorsal attention; RSFC, resting-state functional connectivity; SalVentAttn, salience/ventral attention; SomMot, somatosensory-motor; TempPar, temporoparietal.

(**Figure 5—figure supplement 1**). We note that the second gradient (contrasting somatosensory/motor and visual areas) explained almost the same amount of variance as the first gradient (i.e., 26%, see **Figure 5a**), which may suggest that participants in our sample have only recently transitioned toward a more mature functional organization, that is, more spatially distributed (**Dong et al., 2021**). Next, we tested whether imaging loadings associated to the LCs would follow this sensory-to-transmodal axis (**Sydnor et al., 2021**) by assessing spatial correspondence while adjusting for spatial autocorrelations via spin permutation tests (**Alexander-Bloch et al., 2018**) (see **Supplementary file 1d**). Between-network RSFC loadings for LC1 and LC3 showed a strong positive correlation to the principal gradient (LC1: $r = 0.43$, $p_{spin} < 0.001$; LC3: $r = 0.33$, $p_{spin} < 0.001$; **Figure 5d**), that is, between-network connectivity was higher in transmodal regions and lower in sensory areas. Within-network RSFC loadings for LC1, LC2, and LC3 also showed a significant albeit weak correlation with the sensory-to-transmodal gradient LC1: $r = -0.14$, $p_{spin} = 0.010$; LC2: $r = 0.13$, $p_{spin} = 0.032$; LC3: $r = 0.17$, $p_{spin} = 0.017$ (**Figure 5d**), that is, within-network FC was higher in sensory regions in LC1, and in higher-order regions in LCs 2–3. A similar pattern of findings was observed when cross-validating between- and within-network RSFC loadings to an RSFC gradient derived from an independent dataset (HCP), with strongest correlations seen for between-network RSFC loadings for LC1 and LC3 (LC1: $r=0.50$, $p_{spin} < 0.001$; LC3: $r = 0.37$, $p_{spin} < 0.001$). Of note, we obtained similar correlations when using T1w/T2w ratio in the same participants, a proxy of intracortical microstructure and hierarchy (**Glasser and Van Essen, 2011**). Specifically, we observed the strongest association between this microstructural marker of the cortical hierarchy and between-network RSFC loadings related to LC1 ($r = -0.43$, $p_{spin} < 0.001$). None of the structural loadings were associated with principal gradient scores.

## Contextualization with respect to white matter architecture and across cognitive states

Our final analysis related out findings to white matter architecture and assessed the stability of functional organization across three cognitive states including reward, (emotional) working memory, and impulsivity (**Figure 6**, also see **Figure 2—figure supplement 3** for uncorrected task FC patterns). Greater psychopathology (LC1) was related to widespread decrease in FA and MD in all white matter tracts (see **Supplementary file 1e** for diffusion-based loadings and $Z$-scores for all white matter tracts). Higher internalizing and lower externalizing symptoms (LC2) were associated to higher FA within the left superior corticostriate-frontal and corticostriate tracts, and higher MD within the forceps major and parahippocampal cingulum. Finally, greater neurodevelopmental symptoms (LC3) were related to decreased FA within the right cingulate cingulum, and within the left superior, temporal, and parietal longitudinal fasciculus.

Regarding task FC organization, many patterns appeared to mirror those seen during rest. Greater psychopathology was associated with higher FC between control and default network across all states, including rest (**Figure 6c**). LC1 was also related to higher FC between sensory and attention networks during the Emotional N-back (EN-back task), and decreased FC between control and attention networks during Monetary Incentive Delay (MID) and Stop Signal Task (SST) tasks. LC2 was related to decreased FC within the default network and between the default and control networks across all states, increased FC between somatomotor and control networks during MID and EN-back tasks, and between somatomotor and default network during EN-back and SST tasks. Finally, task FC patterns related to LC3 were similar to those observed during rest, for example, decreased within-network FC and higher FC between default and attention networks.

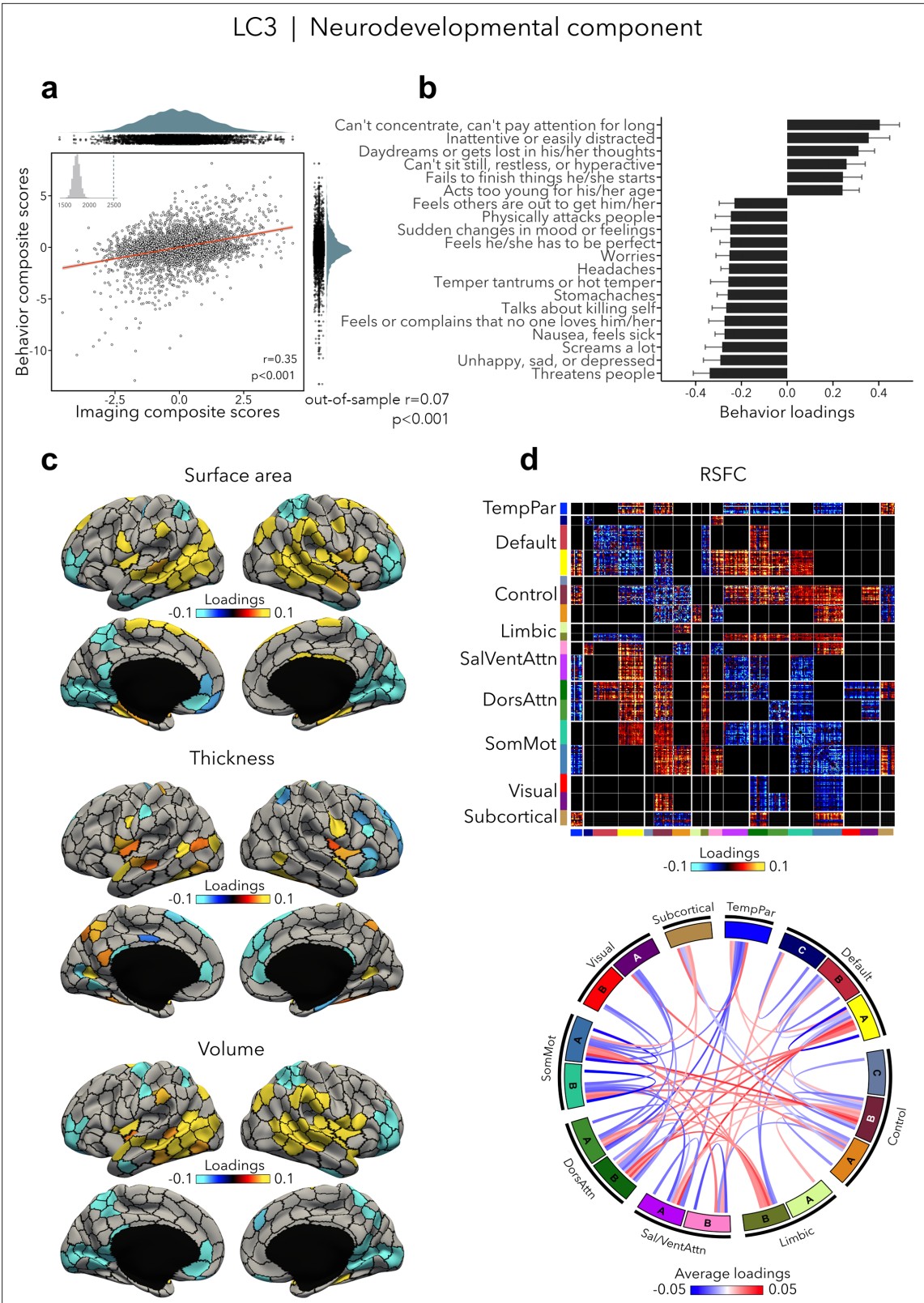

**Figure 4.** Neurodevelopmental component (LC3). (**a**) Correlation between imaging and behavior composite scores (*r* = 0.35, permuted p = 0.001; out-of-sample generalizability of model statistics: *r* = 0.07, permuted p = 0.001). Each dot represents a different participant from the discovery sample (n=3,504). The inset on the top left shows the null distribution of (permuted) singular values from the permutation test, while the dotted line shows the original singular value. (**b**) Top absolute behavior loadings characterizing this component. Higher loadings represent increased (i.e., worse)

*Figure 4 continued on next page*

*Figure 4 continued*

neurodevelopmental symptoms, while lower loadings represent a mix of externalizing and internalizing symptoms linked to emotion dysregulation. Error bars indicate bootstrap-estimated confidence intervals. (**c**) Significant surface area, thickness, and volume loadings (after bootstrap resampling and FDR correction $q < 0.05$) associated with LC3. (**d**) Significant RSFC loadings were thresholded, whereby only within- or between-network blocks with significant bootstrapped Z-scores are shown. Network blocks following the colors associated with the 17 Yeo networks (**Schaefer et al., 2018**; **Yeo et al., 2011**) and subcortical regions (**Fischl et al., 2002**). Chord diagram summarizing significant within- and between-network RSFC loadings. See also **Figure 1a** for more detailed network visualization. DorsAttn, dorsal attention; RSFC, resting-state functional connectivity; SalVentAttn, salience/ventral attention; SomMot, somatosensory-motor; TempPar, temporoparietal.

## Generalizability and control analyses

We implemented different approaches to evaluate the robustness and potential generalizability of our findings. First, we performed a completely independent replication of the analysis pipeline in an unseen sample of participants (see **Figure 7**). We observed significant correlations between behavioral loadings of LCs 1–3 across discovery and replication samples ($r = 0.63$–$0.97$). In terms of imaging loadings, RSFC loadings were replicated in LCs 1–3 ($r = 0.11$–$0.29$, $p_{spin} < 0.05$); thickness loadings were replicated in LCs 1–3 ($r = 0.15$–$0.55$, $p_{spin} < 0.05$); volume loadings were replicated in LCs 1–2 ($r = 0.18$–$0.19$, $p_{spin} < 0.05$) but not LC3 ($r = 0.02$, $p = 0.467$); finally, surface area loadings were only replicated in LC2 ($r = 0.15$, $p_{spin} = 0.040$). However, independently re-calculating model statistics in the replication sample may yield inflated effect sizes in estimating out-of-sample prediction. We addressed this limitation by applying all model weights computed in the discovery sample to the replication sample data. We first applied the imaging principal components analysis (PCA) coefficients computed in the discovery cohort to the raw replication cohort data. Resulting PCA scores and behavioral data were then normalized using the mean and standard deviation of corresponding data in the discovery cohort. Cross-validated composite scores were generated by multiplying singular value decompositions of the discovery cohort data with the normalized imaging PCA and behavioral data from the replication sample. Modality-specific and behavioral loadings were recovered by correlating cross-validated composite scores with normalized replication sample data. With this approach, we found that out-of-sample prediction was overall high across LCs 1–3 for behavioral loading ($r = 0.94$–$0.97$), and lower for imaging loadings ($r = 0.16$–$0.29$). These analyses suggest that questionnaire item loadings were highly replicable across discovery and replication cohorts but indicate lower generalizability of structural and functional network loadings. This lower replicability of brain features also affected out-of-sample prediction statistics linking imaging features and behavior (cross-validated composite scores), which were generally low across LCs but remained statistically significant (LC1 $r = 0.03$; LC2 $r = 0.05$; LC3 $r = 0.07$; all permuted $p < 0.001$ after permuting the first five LCs 10,000 times, accounting for site and FDR).

To make sure that our findings were robust to different PCA thresholds, we repeated our analyses while keeping principal components that explained 10%, 30%, 70%, and 90% of variance in each imaging modality (instead of 50% as for the main analysis). Both behavior and imaging loadings were highly similar to those in our main analysis, though we note that similarity was lower in the control analysis that kept principal components explaining 10% of the data, which is likely due to the very low dimensionality of the data (i.e., 14 principal components) (**Supplementary file 1f**). Third, to account for redundancy within structural imaging metrics included in our main PLS model (i.e., cortical volume is a result of both thickness and surface area), we also repeated our main analysis while excluding cortical volume from our imaging metrics. Findings were very similar to those in our main analysis, with an average absolute correlation of $0.898 \pm 0.114$ across imaging composite scores of LCs 1–5. Considering scan quality in T1w-derived metrics (from manual quality control ratings) yielded similar results to our main analysis, with an average correlation of $0.986 \pm 0.014$ across imaging composite scores. Additionally considering head motion parameters from diffusion imaging metrics in our model yielded consistent results to those in our main (mean $r = 0.891$, SD $= 0.103$; $r = 0.733$–$0.998$). Moreover, repeating the PLS analysis while excluding ethnicity as a model covariate yielded overall similar imaging and behavioral composites scores across LCs to our original analysis. Across LCs 1–5, the average absolute correlations reached $r = 0.636 \pm 0.248$ for imaging composite scores, and $r = 0.715 \pm 0.269$ for behavioral composite scores. Removing these covariates seemed to exert stronger effects on LC3 and LC4 for both imaging and behavior, as lower correlations across models were specifically observed for these components.

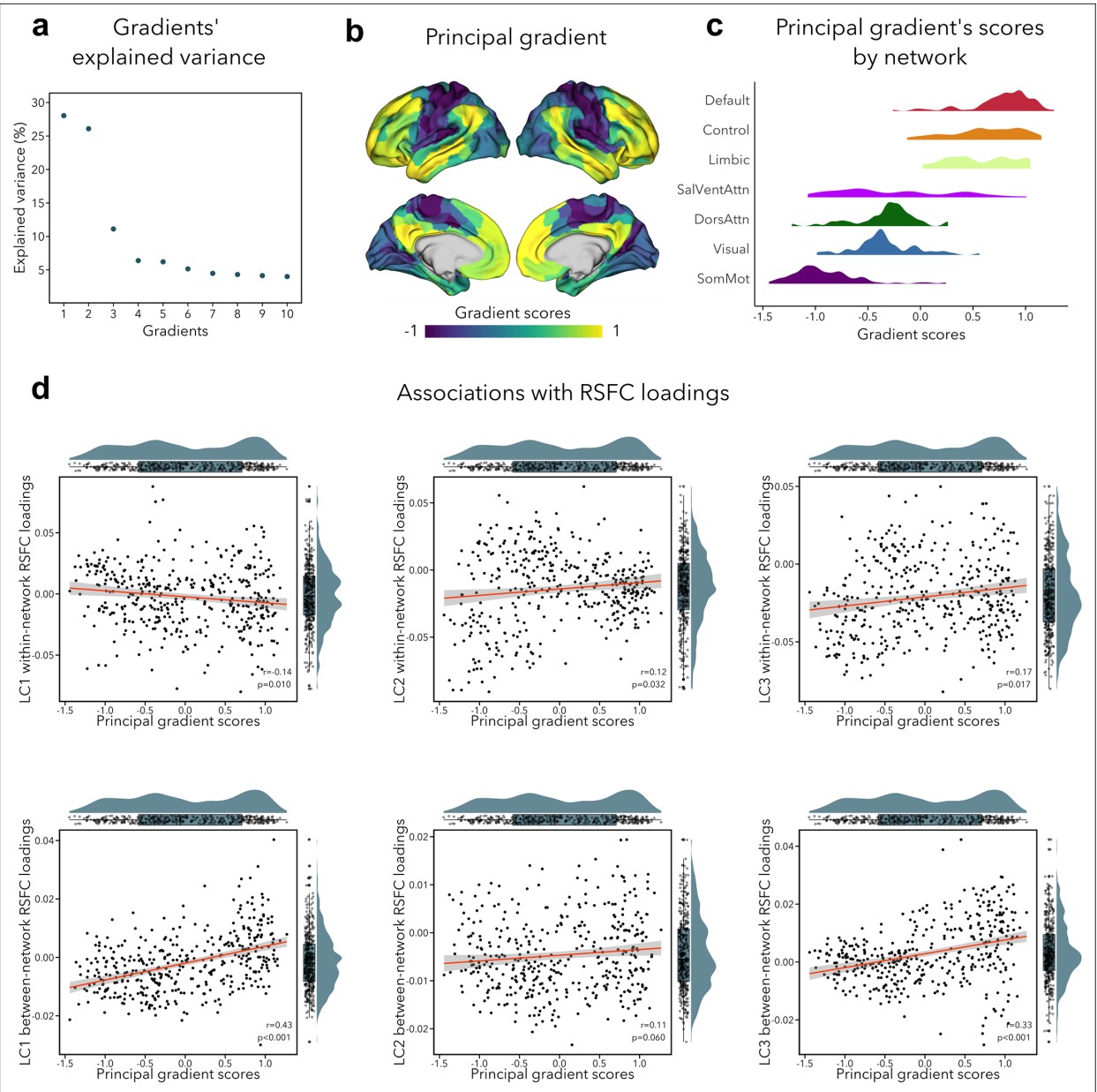

**Figure 5.** Association of functional connectivity loadings with principal functional gradient. (**a**) Percentage of RSFC variance explained by each gradient. (**b**) Principal functional gradient, anchored by transmodal association cortices on one end and by sensory regions on the other end. (**c**) Distribution of principal gradient's scores by cortical network (*Yeo et al., 2011*). (**d**) Associations between principal gradient scores and both within- and between-network RSFC loadings. For all spatial correlations, statistical significance was determined using an autocorrelation-preserving spin permutation procedure (see text, threshold for statistical significance was p < 0.05).

The online version of this article includes the following figure supplement(s) for figure 5:

**Figure supplement 1.** Principal gradient computed without alignment to the gradients derived from the Human Connectome Project (HCP) dataset.

We also explored associations between age/sex and psychopathology dimensions. Notably, we found that male participants had higher composite scores on LC1 (*p* factor) and LC3 (neurodevelopmental symptoms), while female participants had higher composite scores on LC2 (internalizing symptoms) (*Figure 1—figure supplement 2*). Furthermore, both imaging and behavior scores for LC2 (internalizing/externalizing symptoms) were significantly albeit weakly correlated with age and age$^2$ ($r$ = 0.04–0.05, ps < 0.028; *Supplementary file 1g*).

Lastly, although the current work aimed to reduce intrinsic correlations between variables within a given modality through running a PCA before the PLS approach, intrinsic correlations between

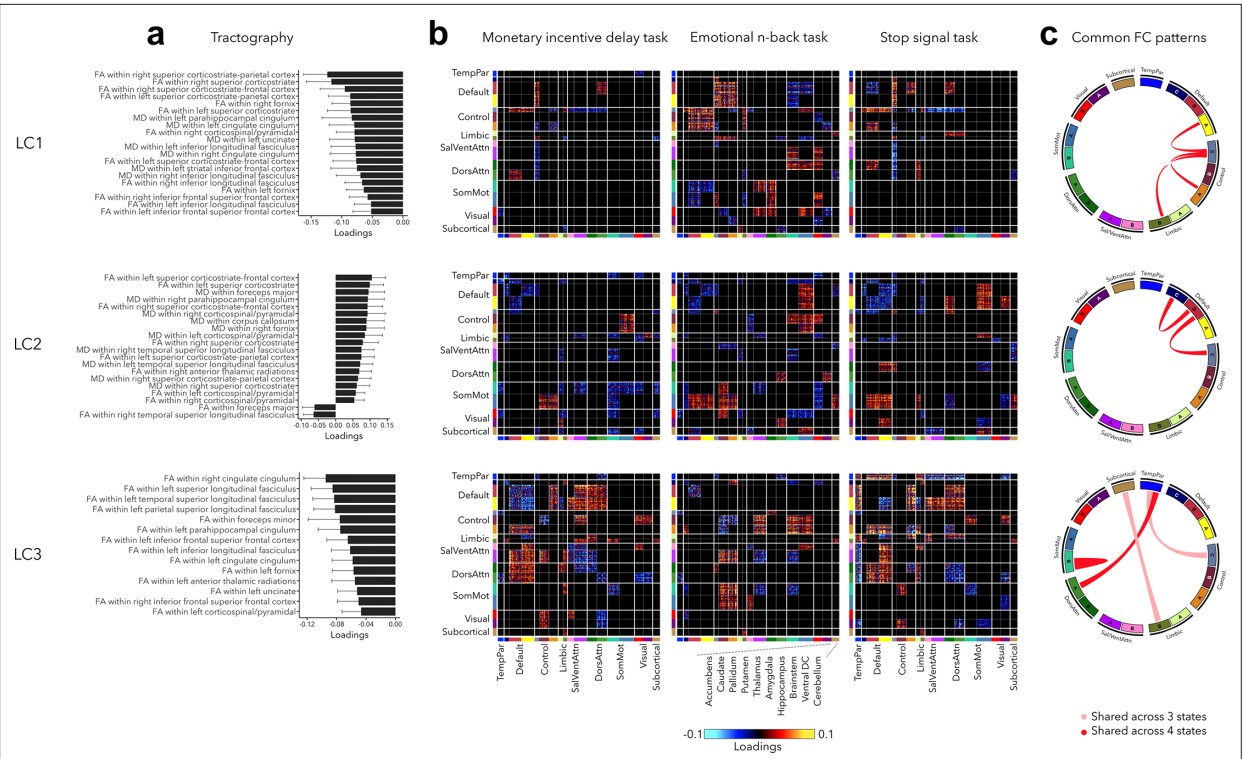

**Figure 6.** Significant diffusion magnetic resonance imaging (MRI) tractography loadings and task FC loadings associated with LC1–LC3, derived in smaller subsamples (*N* = 3275 for tractography and *N* = 1195 for task FC). (**a**) Error bars on the bar charts depicting tractography loadings on the right indicate bootstrap-estimated confidence intervals. (**b**) Task FC loadings were thresholded, whereby only within- or between-network blocks with significant bootstrapped *Z*-scores are shown. Network blocks follow the colors associated with the 17 Yeo networks (*Schaefer et al., 2018*; *Yeo et al., 2011*) and subcortical regions (*Fischl et al., 2002*). (**c**) FC patterns shared across either all three task states or all four states including rest. DorsAttn, dorsal attention; RSFC, resting-state functional connectivity; SalVentAttn, salience/ventral attention; SomMot, somatosensory-motor; TempPar, temporoparietal; Ventral FC, ventral diencephalon.

measures and modalities may potentially be a remaining factor influencing the PLS solution. We thus provided an additional overview of the intrinsic correlations between the different neuroimaging data modalities in the supporting results (*Supplementary file 1c*). We found that volume loadings were correlated with thickness and surface area loadings across all LCs, in line with the expected redundancy of these structural modalities. While between- and within-network RSFC loadings were also significantly correlated across LCs, their associations to structural metrics were more variable.

## Discussion

The multitude of changes during typical and atypical neurodevelopment, and especially those occurring in late childhood and early adolescence, are complex and advocate for multidimensional approaches to comprehensively characterize risk patterns associated with developing mental illness. In the present work, we simultaneously delineated latent dimensions of psychopathology and their structural and functional neural substrates based on a large community-based developmental dataset, the ABCD cohort (*Casey et al., 2018*). Our findings mirrored the psychopathological hierarchy – starting with the *p* factor, followed by decreasingly broad dimensions – suggesting that this hierarchy is represented in multimodal cortical reorganization during development. The *p* factor, internalizing, externalizing, and neurodevelopmental dimensions were each associated with distinct morphological and intrinsic functional connectivity signatures, although these relationships varied in strength. Latent components were also found to scale with initially held out neuroimaging features, including task-based connectivity patterns as well as diffusion MRI metrics sensitive to white matter architecture. Notably, connectivity signatures associated to the different components followed a recently described sensory-to-transmodal axis of cortical organization, which has been suggested to not only

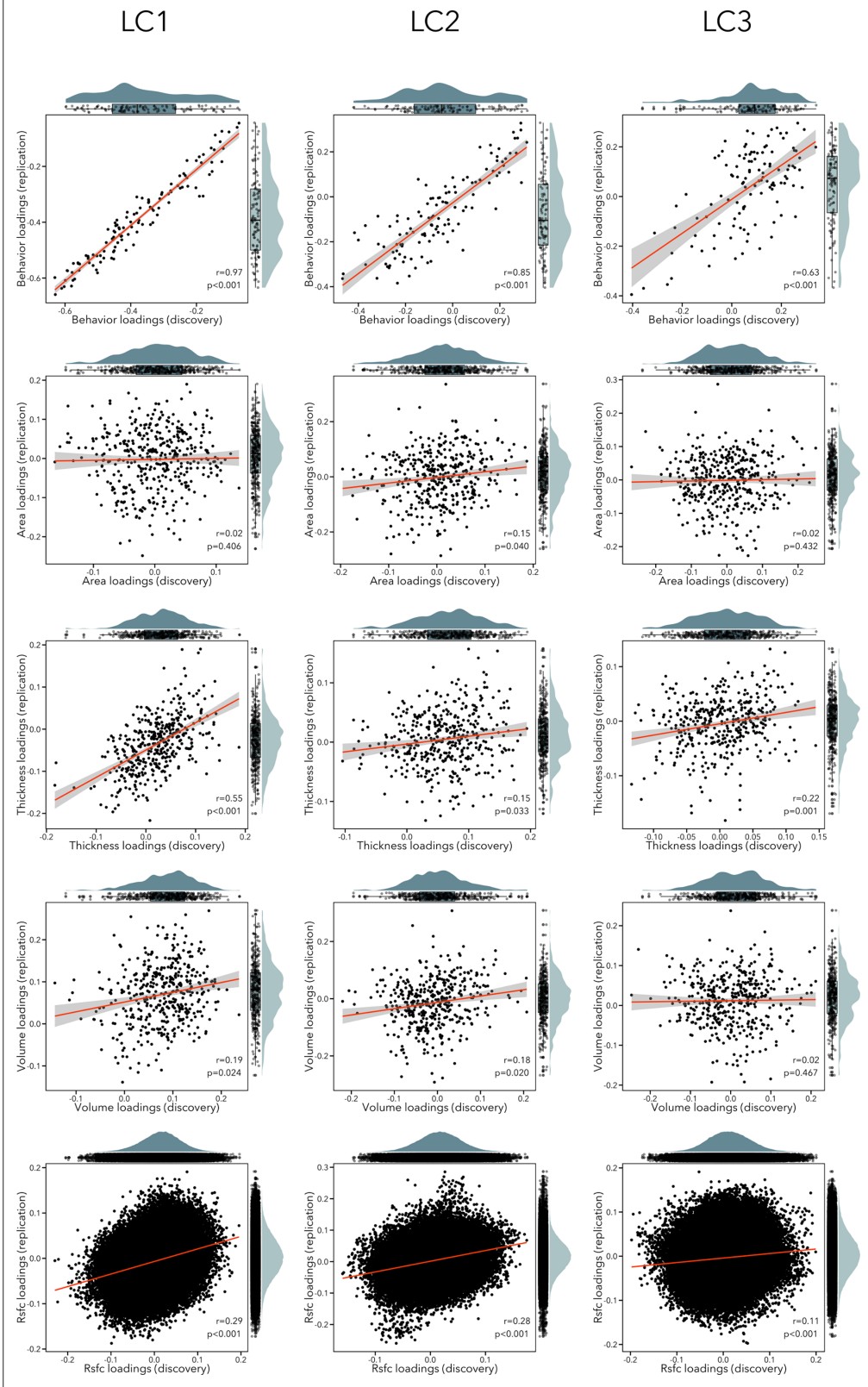

**Figure 7.** Scatterplots showing correlation between loadings in the discovery and replication sample for each modality (rows) and each LC (columns). The distribution of the loadings in the discovery sample are shown on the top *x*-axis, while the distribution of the loadings in the replication sample are depicted on the right *y*-axis.

relate to the emergence of complex cognition and behavior, but also to the potential for neuroplasticity across the cortical landscape as well as risk for psychopathology (*Sydnor et al., 2021*). Finally, our findings were validated in a replication sample and were robust to several parameter variations in analysis methodology. Model generalizability to unseen data was overall low and was likely limited by sample-specific variations in structural and functional imaging features. Although model parameters yielded statistically significant brain–behavior associations in unseen data over LCs 1–3, the poor generalizability of model parameters strongly mitigates the potential of presented neuroimaging signatures to serve screening or diagnostic purposes in detecting childhood psychopathology. Symptom dimensions were consistent with prior literature, but independent replication of the model indicated strong sample-specific variations in structural and functional imaging features which may explain poor generalizability.

Our study combined multiple measures tapping on brain structure that represent biologically meaningful phenotypes capturing distinct evolutionary, genetic, and cellular processes (*Raznahan et al., 2011*). These processes are closely intertwined, and follow a nonlinear (i.e., a curvilinear inverted-U) trajectory during neurodevelopment that peak in late childhood/adolescence (*Raznahan et al., 2011*) – but how and when these processes are disturbed by psychopathology is still poorly understood. We integrated structural neuroimaging features with measures of functional organization at rest, and sought to optimize their covariance with various symptom combinations, extending previous work that operated either on symptomatology (*Michelini et al., 2019*) or neuroimaging features alone (*Elliott et al., 2018*; *Kebets et al., 2019*; *Romer et al., 2021*; *Shanmugan et al., 2016*; *Xia et al., 2018*). Our integrated approach nevertheless replicated the dimensions found by *Michelini et al., 2019*, thereby extending them to a wide range of neurobiological substrates. Moreover, our findings recapitulated the hierarchy within psychopathology (*Kotov et al., 2017*; *Lahey et al., 2017*), with the *p* factor explaining the highest covariance across multiple imaging features, and distinct structural and functional signatures related to internalizing, externalizing, and neurodevelopmental dimensions. Previous research had found that disentangling the variation due to the *p* factor from other dimensions could be challenging, with sometimes few morphological or connectivity changes related to second-order dimensions left after adjusting for the *p* factor (e.g., *Cui et al., 2021*; *Parkes et al., 2021*). By using PLS, which derives orthogonal components (*McIntosh and Lobaugh, 2004*; *McIntosh and Mišić, 2013*), we ensured that the neural substrates associated with LCs 2–5 were independent from those associated with the *p* factor (i.e., LC1). Furthermore, we observed that structural patterns associated with psychopathology dimensions were highly similar across structural modalities, and somewhat similar to functional patterns though to a much lesser extent, which further suggests that structural and functional modalities provide complementary information to characterize vulnerability to psychopathology. While the current work derived main imaging signatures from resting-state fMRI as well as gray matter morphometry, we could nevertheless demonstrate associations to white matter architecture (derived from diffusion MRI tractography) and recover similar dimensions when using task-based fMRI connectivity. Despite subtle variations in the strength of observed associations, the latter finding provided additional support that the different behavioral dimensions of psychopathology more generally relate to alterations in functional connectivity. Given that task-based fMRI data offers numerous avenues for analytical exploration, our findings may motivate follow-up work assessing associations to network- and gradient-based response strength and timing with respect to external stimuli across different functional states.

The *p* factor was associated with widespread decreases in cortical thickness, volume, but also in FA and MD. In line with previous research, this pattern likely reflects a general effect of atypical brain morphology associated with worse overall functioning (*Kaczkurkin et al., 2019*; *Romer et al., 2021*). At the functional level, increased FC between control and default networks was found across all cognitive states – a pattern of dysconnectivity that had previously been reported across dimensions of mood, psychosis, fear, and externalizing behavior (*Xia et al., 2018*). Internalizing and externalizing symptoms were associated with mixed imaging patterns, due to these dimensions being contrasted in a single component characterized by both positive and negative behavior loadings. Cortical thickness patterns were consistent with a recent study reporting increased frontotemporal thickness associated with externalizing behavior in the same cohort (*Modabbernia et al., 2022*). These three broad dimensions (i.e., *p* factor, internalizing, externalizing) are thought to be underpinned by sets of pleiotropic genetic influences that characterize the principal modes of genetic risk transmission for most

disorders in childhood and adolescence (*Pettersson et al., 2016*; *Pettersson et al., 2013*), as well as by environmental events and contextual factors, such as familial situation, trauma, and broader socioeconomic challenges, which may collectively modulate the way these disorders are expressed (*Gur et al., 2019*). The third component resembled the neurodevelopmental dimension, which captures inattention and autistic traits, and has previously been linked to intelligence and academic achievement (*Kim et al., 2018*; *Michelini et al., 2021*; *Michelini et al., 2019*), and more generally a predictor of learning (*Holmes et al., 2021*). At the functional level, the neurodevelopmental component was characterized by decreased within-network RSFC patterns across all cognitive states, in line with previous finding in the same cohort reporting lower RSFC within the default mode network, and altered connectivity of the default and control networks in association with neurodevelopmental symptoms (*Karcher et al., 2021*). Our findings add evidence to this cluster of symptoms having common neurobiological substrates that are distinct from other psychiatric disorders (*Cross-Disorder Group of the Psychiatric Genomics Consortium, 2019*; *Opel et al., 2020*), which is in line with the known broad genetic overlap between neurodevelopmental symptoms, including autistic and ADHD behaviors, as well as learning difficulties, both throughout the general population and at the quantitative extreme (*Pettersson et al., 2013*; *Ronald et al., 2008*). There is also significant phenotypic overlap in the general population between autistic traits and ADHD symptoms, a co-occurrence that might be due to the disruptions in brain development during critical stages in gestation, infancy or early childhood, which could in turn lead to problems that would affect the normal growth process in areas such as learning, social interaction and behavioral control – a process that is primarily genetic in origin (*Ronald et al., 2008*). A recent study found that polygenic risk scores for ADHD and autism were associated with the neurodevelopmental factor in the ABCD cohort, although the latter did not survive after adjusting for the *p* factor (*Waszczuk et al., 2023*).

Conceptual and analytic advances have begun to characterize cortical organization along gradual dimensions, offering a continuous description of cortical arealization and modularity (*Bernhardt et al., 2022*; *Huntenburg et al., 2018*). One major dimension situates large-scale networks along a spectrum running from unimodal regions supporting action and perception to heteromodal association areas implicated in abstract cognition (*Margulies et al., 2016*; *Paquola et al., 2019*). Initially demonstrated at the level of intrinsic functional connectivity (*Margulies et al., 2016*), follow-up work confirmed a similar cortical patterning using microarchitectural in vivo MRI indices related to cortical myelination (*Burt et al., 2018*; *Huntenburg et al., 2017*; *Paquola et al., 2019*), post-mortem cytoarchitecture (*Goulas et al., 2018*; *Paquola et al., 2020*; *Paquola et al., 2019*), or post-mortem microarray gene expression (*Burt et al., 2018*). Spatiotemporal patterns in the formation and maturation of large-scale networks have been found to follow a similar sensory-to-association axis; moreover, there is the emerging view that this framework may offer key insights into brain plasticity and susceptibility to psychopathology (*Sydnor et al., 2021*). In particular, the increased vulnerability of transmodal association cortices in late childhood and early adolescence has been suggested to relate to prolonged maturation and potential for plastic reconfigurations of these systems (*Paquola et al., 2019*; *Park et al., 2022b*). Between mid-childhood and early adolescence, heteromodal association systems such as the default network become progressively more integrated among distant regions, while being more differentiated from spatially adjacent systems, paralleling the development of cognitive control, as well as increasingly abstract and logical thinking. This fine-tuning is underpinned by a gradual differentiation between higher- and lower-order regions, which may be dependent on the maturation of association cortices. As they subserve cognitive, mentalizing, and socioemotional processes, their maturational variability is thought to underpin inter-individual variability in psychosocial functioning and mental illness (*Sydnor et al., 2021*). We found that between-network RSFC loadings related to the *p* factor followed the sensory-to-transmodal gradient, with the default and control networks yielding higher between-network FC (i.e., greater integration) while sensory systems exhibited lower between-network FC (i.e., greater segregation), suggesting that connectivity between the two anchors of the principal gradient might be affected by the *p* factor. This finding is in line with recent studies showing that the sensory-to-transmodal axis is impacted across several disorders (*Hettwer et al., 2022*; *Opel et al., 2020*; *Park et al., 2022a*). As the *p* factor represents a general liability to all common forms of psychopathology, this points toward a disorder-general biomarker of dysconnectivity between lower- and higher-order systems in the cortical hierarchy (*Elliott et al., 2018*; *Kebets et al., 2019*), which might be due to abnormal differentiation between higher- and lower-order brain networks, possibly

reflective of atypical maturation of higher-order networks. Interestingly, a similar pattern, albeit somewhat weaker, was also observed in association with the neurodevelopmental dimension. This suggests that neurodevelopmental difficulties might be related to alterations in various processes orchestrated by sensory and association regions, as well as the macroscale balance and hierarchy of these systems, in line with previous findings in several neurodevelopmental conditions, including autism, schizophrenia, as well as epilepsy, showing a decreased differentiation between the two anchors of this gradient (*Hong et al., 2019*). In future work, it will be important to evaluate these tools for diagnostics and population stratification. In particular, the compact and low-dimensional perspective of gradients may provide beneficial in terms of biomarker reliability as well as phenotypic prediction, as previously demonstrated using typically developing cohorts (*Hong et al., 2020*). On the other hand, it will be of interest to explore in how far alterations in connectivity along sensory-to-transmodal hierarchies provide sufficient granularity to differentiate between specific psychopathologies, or whether they, as the current work suggests, mainly reflect risk for general psychopathology and atypical development.

Our findings should be considered in light of some caveats. First, latent variable approaches such as PLS are powerful methods to characterize modes of covariation between multiple datasets, but they do not inform on any causal associations between them. Second, although they were repeated in a replication cohort, verifying our findings in an independent dataset other than ABCD would indicate broader generalizability. In this regard, our model was found to exhibit relatively poor out-of-sample prediction performance, as is often the case in explorations of complex brain–behavior relationships. This poor generalizability limits the potential of the present findings to meaningfully inform our understanding of the neural basis of psychopathology symptom dimensions and influence clinical practice. Furthermore, we only considered the ABCD cohort's baseline data in our analyses; longitudinal models assessing multiple time points will enable to further model developmental trajectories, test the stability of these dimensions over time, and how they relate to clinical phenotypes (*Brieant et al., 2022*; *Leban, 2021*). Our approach could be expanded to consider brain–environment interactions, as they likely reinforce one another throughout development in shaping different forms of psychopathology (*Sprooten et al., 2022*). For instance, a recent study in the same cohort has shown that a broad range of environmental risk factors, including perinatal complications, socio-demographics, urbanization, and pollution, characterized the main modes of variation in brain imaging phenotypes (*Alnæs et al., 2020*). Although we could consider some socio-demographic variables and proxies of social inequalities relating to race and ethnicity as covariates in our main model, the relationship of these social factors to structural and functional brain phenotypes remains to be established with more targeted analyses. Other factors have also been suggested to impact the development of psychopathology, such as executive functioning deficits (*Zelazo, 2020*), earlier pubertal timing (*Ullsperger and Nikolas, 2017*), negative life events (*Brieant et al., 2021*), maternal depression (*Goodman and Gotlib, 1999*; *Goodman et al., 2011*), or psychological factors (e.g., low effortful control, high neuroticism, and negative affectivity) (*Lynch et al., 2021*). Inclusion of such data could also help to add further insights into the rather synoptic proxy measure of the *p* factor itself (*Fried et al., 2021*), and to potentially assess shared and unique effects of the *p* factor vis-à-vis highly correlated measures of impulse control. Moreover, biases of caregiver reports have been shown with potential divergences between the child's and parent's report depending on family conflict (*Shen et al., 2021*). A large number of missing data in the teachers' reports prevented us from validating these brain–behavior associations with a second caretaker's report. Finally, while prior research has shown that resting-state fMRI networks may be affected by differences in instructions and study paradigm (e.g., with respect to eyes open vs. closed) (*Agcaoglu et al., 2019*), the resting-state fMRI paradigm is homogenized in the ABCD study to be passive viewing of a centrally presented fixation cross. It is nevertheless possible that there were slight variations in compliance and instructions that contributed to differences in associated functional architecture. Notably, however, there is a mounting literature based on high-definition fMRI acquisitions suggesting that functional networks are mainly dominated by common organizational principles and stable individual features, with substantially more modest contributions from task-state variability (*Gratton et al., 2020*). These findings, thus, suggest that resting-state fMRI markers can serve as powerful phenotypes of psychiatric conditions, and potential biomarkers (*Abraham et al., 2017*; *Gratton et al., 2020*; *Parkes et al., 2020*).

Despite these limitations, our study identified several dimensions of psychopathology which recapitulated the psychopathological hierarchy, alongside their structural and functional neural substrates.

These findings are a first step toward capturing multimodal neurobiological changes underpinning broad dimensions of psychopathology, which might be used to predict future psychiatric diagnosis.

## Materials and methods

### Participants

We considered data from 11,875 children from the ABCD 2.0.1 release. The data were collected on 21 sites across the United States (https://abcdstudy.org/contact/), and aimed to be representative of the socio-demographic diversity of the US population of 9–10 year old children (*Garavan et al., 2018*). To ensure that the study had enough statistical power to characterize a large variety of developmental trajectories, the ABCD study aimed for 50% of their sample to exhibit early signs of internalizing/externalizing symptoms (*Garavan et al., 2018*). Ethical review and approval of the protocol was obtained from the Institutional Review Board (IRB) at the University of California, San Diego, as well as from local IRB (*Auchter et al., 2018*). Parents/guardians and children provided written assent (*Clark et al., 2018*). After excluding participants with incomplete structural MRI, resting-state functional MRI (rs-fMRI), or behavioral data, MRI preprocessing, and quality control, and after excluding sites with less than 20 participants, our main analyses included 5251 unrelated children (2577 female [49%], 9.94 ± 0.62 years old, 19 sites). We divided this sample into Discovery (N = 3504, i.e., 2/3 of the dataset) and Replication (N = 1747, i.e., 1/3 of the dataset) subsamples, using randomized data partitioning with both subsamples being matched on age, sex, ethnicity, acquisition site, and overall psychopathology (i.e., scores of the first principal component derived from the 118 items of the Achenbach CBCL *Achenbach and Rescorla, 2013*). *Figure 1d* shows the distribution of these measures in the two samples.

### Behavioral assessment

The parent-reported CBCL (*Achenbach and Rescorla, 2013*) is comprised of 119 items that measure various symptoms in the child's behavior in the past 6 months. Symptoms are rated on a three-point scale from (0 = not true, 1 = somewhat or sometimes true, 2 = very true or always true). We used 118/119 items (see *Supplementary file 1h* for a complete list of items); one item was removed ('Smokes, chews, or sniffs tobacco') as all participants from the discovery sample scored '0' for this question. In the replication sample, another item ('Uses drugs for nonmedical purposes – don't include alcohol or tobacco') was removed for the same reason. Prior to the PLS analysis, effects of age, $age^2$, sex, site, and ethnicity were regressed out from the behavioral and imaging data using a multiple linear regression to ensure that the LCs would not be driven by possible socio-demographic confounders (*Kebets et al., 2021*; *Kebets et al., 2019*; *Xia et al., 2018*). The imaging and behavioral residuals of this procedure were input to the PLS analysis. Of note, the inclusion of ethnicity as a covariate in imaging studies has been recently called into question (*Saragosa-Harris et al., 2022*). In the present study, we included this variable in our main model as a proxy for social inequalities relating to race and ethnicity alongside biological factors (age and sex) with documented effects on brain organization and neurodevelopmental symptomatology queried in the CBCL. We nonetheless quantified the potential effect of this covariate in our main analyses by assessing the consistency of composite scores in a model excluding race and ethnicity covariates (see *Generalizability and control analyses*).

### MRI acquisition

MR images were acquired across 21 sites in the United States with harmonized imaging protocols for GE, Philips, and Siemens scanners (*Casey et al., 2018*). The imaging acquisition protocol consisted of a localizer, T1-weighted images, two runs of rs-fMRI, diffusion-weighted images, T2-weighted images, one to two more runs of rs-fMRI, and the three task-fMRI acquisitions. Full details about the imaging acquisition protocol can be found elsewhere (*Casey et al., 2018*). Scans acquired on the Philips scanner were excluded due to incorrect processing, as recommended by the ABCD consortium.

T1-weighted (T1w) images were acquired using a 3D sequence (1 mm isotropic, Repetition time [TR] = 2500 ms, Echo time [TE] = 2.88 ms, Inversion time [TI] = 1060 ms, flip angle = 8°, matrix = 256 × 256, Field of view [FOV] = 256 × 256 $mm^2$, 176 axial slices) on the Siemens Prisma scanner. Almost identical parameters were used on the GE 750 scanner (except for TE = 2 ms, 208 axial slices). As head

motion is an important concern for (pediatric) imaging, real-time motion detection and correction were implemented for the structural scans and for rs-fMRI at the Siemens sites.

The rs- and task-fMRI data were acquired using a multiband echo planar imaging (EPI) sequence (2.4 mm isotropic voxels, TR = 800 ms, TE = 30 ms, flip angle = 52°, slice acceleration factor 6, matrix = 90 × 90, FOV = 216 × 216 mm$^2$, 60 axial slices) with fast integrated distortion correction. Twenty minutes of rs-fMRI data were acquired in four runs, and participants were instructed to keep their eyes open while passively watching a cross hair on the screen. The three fMRI tasks included an MID task, which measures domains of reward processing, an EN-back task which engages memory and emotion processing, and an SST that engages impulsivity and impulse control. Details about the tasks paradigms and conditions can be found elsewhere (*Casey et al., 2018*). For each participant, there were two runs for each fMRI task.

Diffusion MR images were acquired using a multiband EPI sequence (slice acceleration factor 3) and included 96 diffusion directions, seven $b$ = 0 frames, and four $b$-values (6 directions with $b$ = 500 s/mm$^2$, 15 directions with $b$ = 1000 s/mm$^2$, 15 directions with $b$ = 2000 s/mm$^2$, and 60 directions with $b$ = 3000 s/mm$^2$). Acquisition parameters were almost identical between Siemens (TR = 4100 ms, TE = 88 ms, flip angle = 90°, matrix = 140 × 140, FOV = 240 × 240 mm$^2$, 81 axial slices) and GE 750 scanners (TR = 4100 ms, TE = 81.9 ms, flip angle = 77°, matrix = 140 × 140, FOV = 240 × 240 mm$^2$, 81 axial slices).

## MRI processing

We used minimally preprocessed T1w, fMRI, and diffusion MRI data (*Hagler et al., 2019*). The processing steps for each imaging modality are detailed below: (a) *Structural MRI processing.* T1w images underwent gradient warp correction, bias field correction, and were resampled to a reference brain in standard space with isotropic voxels (*Hagler et al., 2019*). They were further processed using FreeSurfer 5.3.0 (*Dale et al., 1999*; *Fischl et al., 1999a*; *Fischl et al., 1999b*; *Ségonne et al., 2004*; *Ségonne et al., 2007*). Cortical surface meshes were generated for each participant, and registered to a common spherical coordinate system (*Fischl et al., 1999a*, *Fischl et al., 1999b*). Participants that did not pass recon-all quality control were excluded. (b) *fMRI processing.* The same processing was applied to both rs- and task-fMRI data. ABCD initial processing included motion correction, B0 distortion correction, grad warp correction for distortions due to gradient nonlinearities, and resampling to an isotropic resolution. fMRI data further underwent: (1) removal of initial frames *Hagler et al., 2019*; (2) alignment of structural and functional images using boundary-based registration (*Greve and Fischl, 2009*). Runs with boundary-based registration cost >0.6 were excluded. Framewise displacement (FD) (*Jenkinson et al., 2002*) and voxel-wise differentiated signal variance (DVARS) (*Power et al., 2012*) were computed using *fsl_motion_outliers*. Frames with FD >0.3 mm or DVARS >50, along with one frame before and two frames after, were considered as outliers and subsequently censored. Uncensored segments of data fewer than five contiguous frames were also censored (*Gordon et al., 2016*; *Kong et al., 2019*). Runs with >50% frames censored and/or max FD >5 mm were excluded. Participants with less than 4 min of data were also removed. Nuisance covariates including global signal, six motion parameters, averaged ventricular and white matter signal, along with their temporal derivates, were regressed out of the fMRI time series. Censored frames were not considered in the regression. Data were interpolated across censored frames using least squares spectral estimation (*Power et al., 2014*). A bandpass filter (0.009–0.08 Hz) was applied. Finally, preprocessed time series were projected onto FreeSurfer fsaverage6 surface space and smoothed using a 6-mm full-width half maximum kernel. (c) *Diffusion MRI processing.* Initial processing included eddy current distortion, motion correction, B0 distortion correction, grad warp correction, and resampling to an isotropic resolution. Major white matter tracts were labeled using AtlasTrack, a probabilistic atlas-based method for automated segmentation (*Hagler et al., 2009*). Structural MRI images were nonlinearly registered to the atlas and diffusion MRI-derived diffusion orientation for each participant were compared to the atlas fiber orientations, refining a priori tract location probabilities, individualizing the fiber tract ROIs, and minimizing the contribution from regions inconsistent with the atlas (*Hagler et al., 2019*). Processed diffusion MRI data were available in 10,186 participants, among which 3275 overlapped with the Discovery sample (93%).

## Extraction of functional and structural features

RSFC was computed as the Pearson's correlation between the average timeseries among 400 cortical (*Schaefer et al., 2018*) and 19 subcortical (*Fischl et al., 2002*) regions (*Figure 1a*), yielding 87,571 connections for each participant. Censored frames were not considered when computing FC. Age, age$^2$, sex, site, ethnicity, head motion (mean FD), and image intensity (mean DVARS) were further regressed out from the RSFC data.

Surface area, thickness, and volume were extracted from the same 400 cortical regions (*Schaefer et al., 2018*). Age, age$^2$, sex, site, and ethnicity were also regressed out from each parcel-wise structural measure; cortical thickness and volume measures were additionally adjusted for total intracranial volume, and surface area additionally for total surface area.

To reduce data dimensionality before combining the different imaging modalities, we applied PCA over each feature (i.e., surface area, cortical thickness, cortical volume, and RSFC), and selected PCA scores of the number of components explaining 50% of the variance within each data modality, before concatenating them (see *Figure 1b*). The chosen 50% threshold sought to balance the relative contribution of modalities – to prevent the relatively larger number of RSFC features (compared to structural features) from overpowering the analyses; however, we also report results for different thresholds (see *Generalizability and control analyses*). We obtained 50, 58, 57, and 256 principal components for surface area, thickness, volume, and RSFC, respectively, resulting in 421 components in total.

The relative importance of each imaging modality for LCs 1–5 is shown in *Figure 1—figure supplement 3*. To determine the relative contribution of each imaging data modality to the imaging loadings associated with each LC, we computed Pearson's correlations between the 'full' imaging composite scores ($X_{pca} \times V$) and the modality-specific imaging composites scores ($X_{pca} \times V_{modality}$). Modality-specific imaging composite scores were computed by multiplying the imaging data ($X_{pca}$) by the imaging saliences ($V$) after turning into zero all the values of the other modalities and keeping only the saliences of that specific modality. For example, when computing the relative importance of surface area loadings, only the first 50 rows of $V$ are kept (i.e., the number of surface area features kept after PCA), while the rest of the rows are replaced by zeros.

## PLS analysis

PLS correlation analysis (*McIntosh and Lobaugh, 2004*; *McIntosh and Mišić, 2013*) was used to identify *latent components* (LCs) that optimally related children's symptoms (indexed by the CBCL) to structural and functional imaging features (*Figure 1c*). PLS is a multivariate data-driven statistical technique that aims to maximize the covariance between two data matrices by linearly projecting the behavioral and imaging data into a low-dimensional space.

The PLS analysis was computed as follows. The imaging and behavior data are stored in matrices $X_{pca}$ (participants × principal component scores from multimodal imaging data) and $Y$ (participants × CBCL items), respectively. After $Z$-scoring $X_{pca}$ and $Y$ (across all participants), we computed the covariance matrix $R$:

$$R = Y^T \times X_{pca}$$

followed by singular value decomposition of $R$:

$$R = U \times S \times V^T$$

which resulted in three low-dimensional matrices: $U$ and $V$ are the singular vectors comprised of behavioral and imaging *weights* (akin to coefficients in PCA), while $S$ is a diagonal matrix comprised of the singular values. Next, we computed $L_X$ and $L_Y$ by projecting $X$ and $Y$ onto their respective weights $V$ and $U$:

$$L_X = X_{pca} \times V$$
$$L_Y = Y \times U$$

The matrices obtained are the imaging and behavioral *composite scores*, and reflect the participants' individual imaging and behavioral contribution to each LC (akin to factor scores in PCA). The contribution of each variable to the LCs was determined by computing Pearson's correlations between participants' composite scores and their original data, which we refer to as *loadings*. The covariance

explained by each LC was computed as the squared singular value divided by the squared sum of all singular values. Statistical significance of the LCs was assessed using permutation testing (10,000 permutations accounting for site) over the singular values of the first five LCs, while accounting for acquisition site (i.e., data were permuted between participants from the same site). Loading stability was determined using bootstraps, whereby data were sampled 1000 times with replacement among participants from the same site. Bootstrapped Z-scores were computed by dividing each loading by its bootstrapped standard deviation. To limit the number of multiple comparisons, the bootstrapped rs- and task FC loadings were averaged across edge pairs within and between 18 networks, before computing Z-scores. The bootstrapped Z-scores were converted to p-values and FDR-corrected ($q <$ 0.05) along with other posthoc tests. The procedure was performed in both the discovery and replication samples.

## Associations with cortical organization

To map psychopathology-related structural and functional abnormalities along the sensory-to-transmodal gradient of brain organization, we applied diffusion map embedding (*Coifman et al., 2005*), a nonlinear dimensionality reduction technique to the RSFC data. Essentially, strongly inter-connected parcels will be closer together in this low-dimensional embedding space (i.e., have more similar scores), while parcels with little or no inter-covariance will be further apart (and have more dissimilar scores). Previous work has shown that spatial gradients of RSFC variations derived from nonlinear dimensionality reduction recapitulate the putative cortical hierarchy (*Bernhardt et al., 2022*; *Margulies et al., 2016*; *Mesulam, 1998*), suggesting that functional gradients might approximate an inherent coordinate system of the human cortex. To derive the functional gradient, we first calculated a cosine similarity matrix from the average RSFC matrix of our full sample (i.e., discovery and replication samples together, $N = 5,251$). This matrix, thus, captures similarity in connectivity patterns for each pair of regions. Following prior studies (*Margulies et al., 2016*), the RSFC was initially thresholded to only contain the top 10% entries for each row, that is, the 10% strongest connections of each region. The $\alpha$ parameter (set at $\alpha = 0.5$) controls the influence of the density of sampling points on the manifold ($\alpha = 0$, maximal influence; $\alpha = 1$, no influence), while the $t$ parameter (set at $t = 0$) scales eigenvalues of the diffusion operator. These parameters were set to retain the global relations between data points in the embedded space, following prior applications (*Hong et al., 2019*; *Margulies et al., 2016*; *Paquola et al., 2019*). To facilitate comparison with previous work (*Margulies et al., 2016*), the connectivity gradient we derived from our ABCD sample was aligned to gradients derived from the HCP healthy young adult dataset, available in the BrainSpace toolbox (*de Wael et al., 2020*), using Procrustes alignment. In a control analysis, we also computed connectivity gradients without aligning them to the HCP dataset to verify whether the gradients' order would change, as a recent study has shown that the principal gradient transitioned from the somatosensory/motor-to-visual to the sensory-to-transmodal gradient between childhood and adolescence (*Dong et al., 2021*). Finally, for gradient-based contextualization of our findings, we computed Pearson's correlations between cortical PLS loadings (within each imaging modality) and scores from the first gradient (i.e., recapitulating the sensory-to-transmodal axis of cortical organization). Statistical significance of spatial associations was assessed using 1000 spin tests that control for spatial autocorrelations (*Alexander-Bloch et al., 2018*), followed by FDR correction for multiple comparisons ($q < 0.05$).

## Associations with task FC and diffusion imaging data

Task FC and diffusion tensor metrics were also explored in a post hoc association analyses in smaller subsamples. Following the same pipeline as for the rs-fMRI data, FC was computed across the entire timecourse of each fMRI task (thereby capturing both the active task and rest conditions), that is, the MID task, which measures domains of reward processing, the EN-back task, which evaluates memory and emotion processing, and the SST, which engages impulsivity and impulse control. Details about task paradigms and conditions can be found elsewhere (*Casey et al., 2018*). After excluding subjects that did not pass both rs- and task-fMRI quality control, MID and SST data were available in 2039 participants, while EN-back data were available in 3435 participants. A total of 1195 of participants overlapped across the three tasks and the discovery sample (39%). Task FC data were corrected for the same confounds as RSFC (i.e., age, age$^2$, sex, site, ethnicity, mean FD, and mean DVARS). Diffusion tensor metrics included FA and MD in 35 white matter tracts (*Hagler et al., 2009*). FA

measures directionally constrained diffusion of water molecules within the white matter and MD the overall diffusivity, and both metrics have been suggested to index fiber architecture and microstructure. Effects of age, $age^2$, sex, site, and ethnicity were regressed out from the FA and MD measures. We tested another model which additionally included head motion parameters as regressors in our analyses of FA and MD measures and assessed the consistency of findings from both models.

The contribution of task FC and diffusion MRI features was computed by correlating participants' task FC and diffusion MRI data with their imaging and behavior composite scores (from the main PLS analysis using structural and RSFC features). As for other modalities, the loadings' stability was determined via bootstraps (i.e., 1000 samples with replacement accounting for site).

## Generalizability and control analyses

Several analyses assessed reliability. First, we repeated the PLS analysis in the replication sample ($N$ = 1,747, i.e., 1/3 of our sample), and tested the reliability of our findings by computing Pearson's correlations between the obtained behavior/imaging loadings with the original loadings. We also assessed the generalizability of our findings by applying model weights computed in the discovery sample to the replication sample data. Second, we repeated the PLS analyses while keeping principal components explaining 10–90% of the variance within each imaging modality, and compared resulting loadings to those of our original model (which kept principal components explaining 50% of the variance) via Pearson's correlation between loadings. For imaging loadings, we computed correlations for each imaging modality, then averaged correlations across all imaging modalities. As cortical volume is a result of both thickness and surface area, we repeated our main PLS analysis while excluding cortical volume from our imaging metrics and report the consistency of these findings with our main model. We also considered manual quality control ratings as a measure of T1w scan quality. This metric was included as a covariate in a multiple linear regression model accounting for potential confounds in the structural imaging data, in addition to age, $age^2$, sex, site, ethnicity, intracranial volume (ICV), and total surface area. Downstream PLS results were then benchmarked against those obtained from our main model.

We also further assessed the effects of socio-demographic profiles of our participant sample. Effects of age and sex differences on the LCs were assessed by computing associations between participants' imaging/behavior composite scores and their age and sex, using either $t$-tests (for associations with sex) or Pearson's correlations (for associations with age). Post hoc tests were corrected for multiple comparisons using FDR correction ($q < 0.05$). We also assessed the replicability of our findings when removing race and ethnicity covariates prior to computing the PLS analysis and correlating imaging and behavioral composite scores across both models.

## Acknowledgements

JR receives financial support from the Canadian Institutes of Health Research (CIHR). VK acknowledges postdoctoral training support by the Transforming Autism Care Consortium (TACC) and the Montreal Neurological Institute (MNI). BTTY is supported by the NUS Yong Loo Lin School of Medicine (NUHSRO/2020/124/TMR/LOA), the Singapore National Medical Research Council (NMRC) LCG (OFLCG19May-0035), NMRC CTG-IIT (CTGIIT23jan-0001), NMRC OF-IRG (OFIRG24jan-0030), NMRC STaR (STaR20nov-0003), Singapore Ministry of Health (MOH) Centre Grant (CG21APR1009), the Temasek Foundation (TF2223-IMH-01), and the United States National Institutes of Health (R01MH133334). Our computational work was partially performed on resources of the National Supercomputing Centre, Singapore (https://www.nscc.sg). Any opinions, findings, and conclusions or recommendations expressed in this material are those of the authors and do not reflect the views of the funders. BB acknowledges research support from the Natural Sciences and Engineering Research Council of Canada (NSERC Discovery-1304413), the CIHR (FDN-154298, PJT-174995), SickKids Foundation (NI17-039), Azrieli Center for Autism Research (ACAR-TACC), BrainCanada, Healthy Brains and Healthy Lives, and the Tier-2 Canada Research Chairs program. Data used in the preparation of this article were obtained from the Adolescent Brain Cognitive Development (ABCD) Study (https://abcdstudy.org), held in the NIMH Data Archive (NDA). This is a multisite, longitudinal study designed to recruit more than 10,000 children age 9–10 and follow them over 10 years into early adulthood. The ABCD Study is supported by the National Institutes of Health and additional federal partners under award numbers U01DA041048, U01DA050989, U01DA051016,

U01DA041022, U01DA051018, U01DA051037, U01DA050987, U01DA041174, U01DA041106, U01DA041117, U01DA041028, U01DA041134, U01DA050988, U01DA051039, U01DA041156, U01DA041025, U01DA041120, U01DA051038, U01DA041148, U01DA041093, U01DA041089, U24DA041123, and U24DA041147. A full list of supporters is available at https://abcdstudy.org/federal-partners.html. A listing of participating sites and a complete listing of the study investigators can be found at https://abcdstudy.org/consortium_members/. ABCD consortium investigators designed and implemented the study and/or provided data but did not necessarily participate in the analysis or writing of this report. This manuscript reflects the views of the authors and may not reflect the opinions or views of the NIH or ABCD consortium investigators. The ABCD data repository grows and changes over time. The ABCD data used in this report came from http://dx.doi.org/10.15154/1504041.

## Additional information

### Funding

| Funder | Grant reference number | Author |
| --- | --- | --- |
| NUS Yong Loo Lin School of Medicine | NUHSRO/2020/124/TMR/LOA | BT Thomas Yeo |
| Singapore National Medical Research Council | OFLCG19May-0035 | BT Thomas Yeo |
| Singapore Ministry of Health | CG21APR1009 | BT Thomas Yeo |
| Temasek Foundation | TF2223-IMH-01 | BT Thomas Yeo |
| National Institutes of Health | R01MH133334 | BT Thomas Yeo |
| Natural Sciences and Engineering Research Council of Canada | NSERC Discovery-1304413 | Boris C Bernhardt |
| Canadian Institutes of Health Research | FDN-154298 | Boris C Bernhardt |
| Sick Kids Foundation | NI17-039 | Boris C Bernhardt |
| Singapore National Medical Research Council | CTGIIT23jan-0001 | BT Thomas Yeo |
| Singapore National Medical Research Council | OFIRG24jan-0030 | BT Thomas Yeo |
| Singapore National Medical Research Council | STaR20nov-0003 | BT Thomas Yeo |
| Canadian Institutes of Health Research | PJT-174995 | Boris C Bernhardt |

The funders had no role in study design, data collection, and interpretation, or the decision to submit the work for publication.

### Author contributions

Jessica Royer, Formal analysis, Validation, Investigation, Writing – review and editing; Valeria Kebets, Conceptualization, Resources, Data curation, Software, Formal analysis, Funding acquisition, Validation, Investigation, Visualization, Methodology, Writing - original draft, Project administration, Writing – review and editing; Camille Piguet, Conceptualization, Writing – review and editing; Jianzhong Chen, Data curation, Formal analysis, Writing – review and editing; Leon Qi Rong Ooi, Data curation, Software, Methodology; Matthias Kirschner, Writing – review and editing; Vanessa Siffredi, Bratislav Misic, Methodology, Writing – review and editing; BT Thomas Yeo, Boris C Bernhardt, Conceptualization, Resources, Software, Supervision, Funding acquisition, Methodology, Project administration, Writing – review and editing

### Author ORCIDs
Jessica Royer https://orcid.org/0000-0002-4448-8998
Valeria Kebets https://orcid.org/0000-0003-1707-7437
Bratislav Misic https://orcid.org/0000-0003-0307-2862
BT Thomas Yeo https://orcid.org/0000-0002-0119-3276
Boris C Bernhardt https://orcid.org/0000-0001-9256-6041

### Ethics
Ethical review and approval of the protocol was obtained from the Institutional Review Board (IRB) at the University of California, San Diego, as well as from local IRB (Auchter et al., 2018; https://doi.org/10.1016/j.dcn.2018.04.003). Parents/guardians and children provided written assent (Clark et al., 2018; https://doi.org/10.1016/j.dcn.2017.06.005).

### Decision letter and Author response
Decision letter https://doi.org/10.7554/eLife.87992.sa1
Author response https://doi.org/10.7554/eLife.87992.sa2

## Additional files

### Supplementary files
Supplementary file 1. Supplementary tables and information cited in this study.
MDAR checklist

### Data availability
The ABCD data are publicly available via the NIMH Data Archive. Processed data from this study (including imaging features, PLS loadings, and composite scores) have been uploaded to the NDA. Researchers with access to the ABCD data will be able to download the data here: https://nda.nih.gov/abcd/. The preprocessing pipeline can be found at https://github.com/ThomasYeoLab/CBIG/tree/master/stable_projects/preprocessing/CBIG_fMRI_Preproc2016 (*Aihuiping et al., 2024*). Preprocessing code specific to this study can be found here: https://github.com/ThomasYeoLab/ABCD_scripts (copy archived at *Chen, 2025*). The code for analyses can be found here: https://github.com/valkebets/multimodal_psychopathology_components (copy archived at *Kebets, 2025*).

The following previously published dataset was used:

| Author(s) | Year | Dataset title | Dataset URL | Database and Identifier |
|---|---|---|---|---|
| National Institutes of Health | 2018 | Adolescent Brain Cognitive Development Study (ABCD) | https://doi.org/10.15154/1504041 | NIMH Data Archive, 10.15154/1504041 |

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
