## [Editor Report]

This important study provides evidence for associations between transdiagnostic psychiatric symptom domains and brain structure and function in a large cohort. The evidence supporting the findings is solid in that brain-behaviour associations are validated in separate subsamples of the data, although out-of-sample accuracies are modest. This study will be of broad interest to researchers interested in the neurobiological basis of mental disorders.

---

## [Decision Letter]

**Decision letter after peer review:**

Thank you for submitting your article "Multimodal neural correlates of childhood psychopathology" for consideration by *eLife*. Your article has been reviewed by three peer reviewers, and the evaluation has been overseen by a Reviewing Editor and Jonathan Roiser as the Senior Editor. Two of the three reviewers have agreed to be named: Eugene Duff (Reviewer 1) and Ted Satterthwaite (Reviewer 2).

Essential revisions (for the authors):

This manuscript represents a multifaceted and comprehensive contribution toward the understanding of the link between functional connectivity and latent dimensions of psychopathology. The paper presents an extensive series of analyses integrating multimodal data derived from the ABCD study and results in the estimation and characterisation of multivariate mappings between psychopathology and neurobiology using partial least squares.

There is a consensus amongst the reviewers that this is a relatively well-executed study that provides a contribution suitable for a broad readership. However, the reviewers also highlight several shortcomings that should be addressed before this manuscript can be considered suitable for publication in *eLife*. The most important of these are:

1) A lack of out-of-sample assessment metrics makes the generalisability and effect sizes of the PLS findings uncertain. We recognise that the authors have used a discovery-replication approach and have re-estimated their multivariate regression model in disjoint subsets of the ABCD sample. However, this is not the same as out-of-sample prediction, which the reviewers feel would yield less biased estimates of generalisability.

2) The reviewers also felt that the authors ought to give attention to the possibility that their findings might be influenced by intrinsic correlations between the neuroimaging data modalities rather than anything specific to the latent component they belong to.

3) Please also provide a more nuanced discussion surrounding the use of the p-factor both from a wider theoretical perspective, accommodating discussion in the field and with particular reference to the presented results. The editors agree with the sentiment expressed by the reviewers that the leading PLS component is not convincingly demonstrated to relate to the p-factor.

4) Finally, the reviewers request that the authors give careful attention to the clarity of exposition of the many analyses conducted, including describing how potential confounding factors such as site, family structure, and ethnicity were accommodated during the modelling procedure. This may include adapting the approaches employed to match best practices as appropriate.

*Reviewer #2 (Recommendations for the authors):*

Out-of-sample testing: I was surprised that the authors didn't use their split sample to conduct out-of-sample testing or use cross-validation; this would provide a more robust measure of effect size.

Site/family structure: in ABCD these variables are often accounted for via multilevel models / random effects in a mixed effects model to account for the nested structure of the data.

Anatomical features: providing context (and potentially moving volume to the supplement) may facilitate the interpretation of this result for readers.

Ethnicity: the rationale for regressing ethnicity from the data was unclear and may conflict with current best practices. See https://www.nature.com/articles/s41593-022-01218-y

Data quality: for a relevant paper for ABCD, See: https://www.biorxiv.org/content/10.1101/2023.02.28.530498v1

Including the Euler number (https://www.sciencedirect.com/science/article/pii/S1053811917310832) or the manual ratings from the ABCD preprint would mitigate these concerns. For dMRI data I would suggest including a summary measure of in-scanner motion as a covariate.

*Reviewer #3 (Recommendations for the authors):*

While reading the paper, I noticed that the authors have included a substantial amount of analyses. However, it was not entirely clear to me why certain data were utilized in each analysis. In order to enhance the clarity and comprehension of their work, I recommend the following improvements:

- Provide a concise and explicit description of the data used for each analysis. This will help readers understand the specific datasets employed in each analysis and their relevance to each analytical approach.

- Given that the structural and rsfMRI data were used for the main PLS analysis and the rest for validation, consider providing a more detailed explanation of the rationale behind this choice. Additionally, elucidate how these datasets contribute to the overall findings and conclusions of your study.

- Enhance validation procedures: if I'm correct, all the PLS inferences were done in sample, so I wonder about the transferability of their found results.

Furthermore, to strengthen their argument regarding the relationship between changes in the sensory-to-transmodal axis and behavioral factors, it would be advisable for the authors to directly correlate composite scores reflecting behavior with the gradients. Additionally, if these correlations turn out to be non-significant, it would be interesting for the authors to discuss the possible reasons behind these findings.

Lastly, to address the interpretation concerns for the first latent component, I suggest the following:

- Delve deeper into the observed loadings: Provide a detailed analysis of the specific loadings associated with LC1 and how they relate to impulse control problems over different train-test sets. By exploring the nuances of these loadings, the authors can offer a more precise understanding of the factor's nature and its connection to the broader p-factor construct.

- Acknowledge the ongoing discussion: Recognize the existing discourse within the field regarding the interpretation and utilization of the p-factor. Discuss the differing perspectives and highlight the points of contention, emphasizing the need for further investigation and clarification.

[Editors' note: further revisions were suggested prior to acceptance, as described below.]

Thank you for resubmitting your work entitled "Multimodal neural correlates of childhood psychopathology" for further consideration by *eLife*. Your revised article has been evaluated by Jonathan Roiser (Senior Editor) and a Reviewing Editor, Andre Marquand.

The manuscript has been improved but there are some remaining issues that need to be addressed, as outlined below:

We thank the authors for their revised manuscript and their responses to the points raised by the reviewers. Before moving forward with further consideration of this revised submission, we would like to ask the authors to extend the out-of-sample analysis that has been provided and to report this more transparently because this was identified as a critical point by the reviewers in the last submission.

More specifically, it seems that only the replicability of the in-sample and out-of-sample estimates is currently provided (and not the actual out-of-sample predictions) although this is not very clear from the description given (e.g. lines 388-390). Please report the out-of-sample prediction statistics (i.e. corresponding to the in-sample estimates currently shown in Figure 2A). Please also test the significance of these for example using permutation testing and adjust the Discussion section accordingly.

Additionally, please clarify the exact steps taken during the out-of-sample estimation procedure. As noted above, this is currently quite unclear. The standard approach within machine learning would be to keep the training and test sets completely independent, where any normalisation of the features prior to prediction is performed using statistics derived from the training set.

[Editors' note: further revisions were suggested prior to acceptance, as described below.]

Thank you for resubmitting your work entitled "Multimodal neural correlates of childhood psychopathology" for further consideration by *eLife*. Your revised article has been evaluated by Jonathan Roiser (Senior Editor) and a Reviewing Editor.

The manuscript has been improved but there are some remaining issues that need to be addressed, as outlined below:

The authors have responded satisfactorily to most of the concerns raised by the reviewers. However there is one significant concern that must be addressed before we can consider this suitable for publication in *eLife*. The focus and narrative of the paper is still nearly entirely based around in-sample statistics, especially the canonical correlations reported in figures 2, 3 and 4 (r ~ 0.35). We feel that this is too optimistic and does not accurately reflect the true magnitude of the effects reported, even in view of the discovery-replication conducted. This is because the out-of-sample canonical correlations now (briefly) included in the manuscript are of a much smaller magnitude (r=0.03 – 0.07), indicating a very small amount of explained variance.

We do not consider the low explained variance to be problematic per se, and indeed it is in line with current standards in the literature (e.g. https://www.nature.com/articles/s41586-022-04492-9), but it should be transparently and accurately reported. In general, given the well-established high propensity of CCA/PLS to overfit thus, resulting in quite brittle models especially where large number of predictor variables are included, we consider that in-sample canonical correlations are not appropriate indicators of model performance in neuroimaging and we should rely on out-of-sample statistics instead (see e.g. https://pubmed.ncbi.nlm.nih.gov/32224000/ for a discussion on this). It is perhaps also useful to note that this view is shared by the reviewers and the reviewing editor who assessed this manuscript.

To address this, please: (i) replace the in-sample statistics reported in all the relevant figures with out-of-sample statistics (or simply add the out of sample statistics to the figures), (ii) report the out-of-sample canonical correlations in the abstract, and (iii) adjust narrative of the paper (and where appropriate downstream analyses) accordingly to focus principally on the out-of-sample statistics.

---

## [Author Response]

Essential revisions (for the authors):This manuscript represents a multifaceted and comprehensive contribution toward the understanding of the link between functional connectivity and latent dimensions of psychopathology. The paper presents an extensive series of analyses integrating multimodal data derived from the ABCD study and results in the estimation and characterisation of multivariate mappings between psychopathology and neurobiology using partial least squares.There is a consensus amongst the reviewers that this is a relatively well-executed study that provides a contribution suitable for a broad readership. However, the reviewers also highlight several shortcomings that should be addressed before this manuscript can be considered suitable for publication in eLife. The most important of these are:

We thank the Reviewers for the positive evaluations and thoughtful comments, which we addressed below.

1) A lack of out-of-sample assessment metrics makes the generalisability and effect sizes of the PLS findings uncertain. We recognise that the authors have used a discovery-replication approach and have re-estimated their multivariate regression model in disjoint subsets of the ABCD sample. However, this is not the same as out-of-sample prediction, which the reviewers feel would yield less biased estimates of generalisability.

We thank the Reviewers for this comment, and we agree that out-of-sample prediction indeed provides stronger estimates of generalizability.

We first applied the PCA coefficients derived from the discovery cohort imaging data to the replication cohort imaging data. The resulting PCA scores and behavioral data were then z-scored using the mean and standard deviation of the replication cohort. The SVD weights derived from the discovery cohort were applied to the normalized replication cohort data to derive imaging and behavioral composite scores, which were used to recover the contribution of each imaging and behavioral variable to the LCs (*i.e.,* loadings). Out-of-sample replicability of imaging (mean r=0.681, S.D.=0.131) and behavioral (mean r=0.948, S.D.=0.022) loadings was generally high across LCs 1-5. Please see the revised manuscript (P.18).

“Generalizability of reported findings was also assessed by directly applying PCA coefficients and latent components weights from the PLS analysis performed in the discovery cohort to the replication sample data. Out-of-sample prediction was overall high across LCs1-5 for both imaging (mean r=0.681, S.D.=0.131) and behavioral (mean r=0.948, S.D.=0.022) loadings.”

2) The reviewers also felt that the authors ought to give attention to the possibility that their findings might be influenced by intrinsic correlations between the neuroimaging data modalities rather than anything specific to the latent component they belong to.

While the current work aimed to reduce intrinsic correlations between variables within a given modality through running a PCA before applying PLS, intrinsic correlations between measures and modalities may generally be a concern for multivariate approaches such as PLS. We comment on this possibility in the revised *Discussion*. We also provide cross-variable correlation matrices in the supplementary results of the paper. Please see P.19-20*.*

“Lastly, although the current work aimed to reduce intrinsic correlations between variables within a given modality through running a PCA before the PLS approach, intrinsic correlations between measures and modalities may potentially be a remaining factor influencing the PLS solution. We thus provided an additional overview of the intrinsic correlations between the different neuroimaging data modalities in the supporting results (Supplementary file 1c). We found that volume loadings were correlated with thickness and surface area loadings across all LCs, in line with the expected redundancy of these structural modalities. While between- and within-network RSFC loadings were also significantly correlated across LCs, their associations to structural metrics were more variable.”

3) Please also provide a more nuanced discussion surrounding the use of the p-factor both from a wider theoretical perspective, accommodating discussion in the field and with particular reference to the presented results. The editors agree with the sentiment expressed by the reviewers that the leading PLS component is not convincingly demonstrated to relate to the p-factor.

Our manuscript now nuances the discussion of the association of LC1 to the p factor. We discuss some of the ongoing debate about the use of the p factor, and cite the recommended publication on P.27.

“Other factors have also been suggested to impact the development of psychopathology, such as executive functioning deficits (Zelazo, 2020), earlier pubertal timing (Ullsperger and Nikolas, 2017), negative life events (Brieant et al., 2021), maternal depression (Goodman and Gotlib, 1999; Goodman et al., 2011), or psychological factors (e.g., low effortful control, high neuroticism, negative affectivity) (Lynch et al., 2021). Inclusion of such data could also help to add further insights into the rather synoptic proxy measure of the p factor itself (Fried et al., 2021), and to potentially assess shared and unique effects of the p factor vis-à-vis highly correlated measures of impulse control.”

4) Finally, the reviewers request that the authors give careful attention to the clarity of exposition of the many analyses conducted, including describing how potential confounding factors such as site, family structure, and ethnicity were accommodated during the modelling procedure. This may include adapting the approaches employed to match best practices as appropriate.

We explored several additional model configurations to assess the influence of including different confound combinations in the PLS analysis. This included the implementation of a minimally strict approach which only considered the effects of participant age, age^2^, site, and sex as covariates. We also considered additional confounds, such as head motion for diffusion MRI analyses and quality control ratings for structural imaging, as requested by the Reviewers. The consistency of our findings was also assessed when excluding one of the structural imaging features (volume), as suggested by a Reviewer. In all cases, composite scores from these additional models were benchmarked against those from the main model described in our manuscript. Model variations overall yielded very similar results, and their consistency with our original findings as well as rare discrepancies are reported and discussed in the revised manuscript.

Reviewer #2 (Recommendations for the authors):Out-of-sample testing: I was surprised that the authors didn't use their split sample to conduct out-of-sample testing or use cross-validation; this would provide a more robust measure of effect size.

As discussed in the editorial summary of essential revisions, we agree that out-of-sample prediction indeed provides stronger estimates of generalizability. We assess this by applying the PCA coefficients derived from the discovery cohort imaging data to the replication cohort imaging data. The resulting PCA scores and behavioral data were then z-scored using the mean and standard deviation of the replication cohort. The SVD weights derived from the discovery cohort were applied to the normalized replication cohort data to derive imaging and behavioral composite scores, which were used to recover the contribution of each imaging and behavioral variable to the LCs (*i.e.,* loadings). Out-of-sample replicability of imaging (mean r=0.681, S.D.=0.131) and behavioral (mean r=0.948, S.D.=0.022) loadings was generally high across LCs 1-5. This analysis is reported in the revised manuscript (P.18).

“Generalizability of reported findings was also assessed by directly applying PCA coefficients and latent components weights from the PLS analysis performed in the discovery cohort to the replication sample data. Out-of-sample prediction was overall high across LCs1-5 for both imaging (mean r=0.681, S.D.=0.131) and behavioral (mean r=0.948, S.D.=0.022) loadings.”

Site/family structure: in ABCD these variables are often accounted for via multilevel models / random effects in a mixed effects model to account for the nested structure of the data.

We included site as a covariate in our linear model, and only included unrelated participants in all analyses (*i.e.*, in both discovery and replication samples).

Anatomical features: providing context (and potentially moving volume to the supplement) may facilitate the interpretation of this result for readers.

We addressed this point by rerunning the PLS analysis while only including thickness and surface area as our structural metrics, to account for the redundancy of these measures with volume. We reduced the reporting of correlations between the loadings from the different modalities in the revised Results (specifically subsections on LC1, LC2, and LC3). Instead, we now refer to Table S4 in each subsection for this information: “Spatial correlations between modality-specific loadings are reported in Supplementary file 1c.”

We also reran the PLS analysis while only including thickness and surface area as our structural metrics, to account for potential redundancy of these measures with volume. This analysis and associated findings are reported on P.36 and P.19:

“As cortical volume is a result of both thickness and surface area, we repeated our main PLS analysis while excluding cortical volume from our imaging metrics and report the consistency of these findings with our main model.”

“Third, to account for redundancy within structural imaging metrics included in our main PLS model (i.e., cortical volume is a result of both thickness and surface area), we also repeated our main analysis while excluding cortical volume from our imaging metrics. Findings were very similar to those in our main analysis, with an average absolute correlation of 0.898±0.114 across imaging composite scores of LCs 1-5.”

Ethnicity: the rationale for regressing ethnicity from the data was unclear and may conflict with current best practices. See https://www.nature.com/articles/s41593-022-01218-y

The rationale for including ethnicity as a covariate in our main model has been clarified in the revised manuscript. For completeness, we also described the results of an additional model with minimal covariates, which notably does not include ethnicity as a covariate. Results of this model were generally similar to our main findings, although some components (particularly LC3 and LC4) showed relatively lower replicability when removing this covariate. We discussed these findings in the revised manuscript.

In light of recent discussions on including this covariate in large datasets such as ABCD (e.g., Saragosa-Harris et al., 2022), we elaborate on our rationale for including this variable in our model in the revised manuscript on P.30:

“Of note, the inclusion of ethnicity as a covariate in imaging studies has been recently called into question. In the present study, we included this variable in our main model as a proxy for social inequalities relating to race and ethnicity alongside biological factors (age, sex) with documented effects on brain organization and neurodevelopmental symptomatology queried in the CBCL.”

We also assess the replicability of our analyses when removing race and ethnicity covariates prior to computing the PLS analysis and correlating imaging and behavioral composite scores across both models. We report resulting correlations in the revised manuscript (P.37, 19, and 27):

“We also assessed the replicability of our findings when removing race and ethnicity covariates prior to computing the PLS analysis and correlating imaging and behavioral composite scores across both models.”

“Moreover, repeating the PLS analysis while excluding this variable as a model covariate yielded overall similar imaging and behavioral composites scores across LCs to our original analysis. Across LCs 1-5, the average absolute correlations reached r=0.636±0.248 for imaging composite scores, and r=0.715±0.269 for behavioral composite scores. Removing these covariates seemed to exert stronger effects on LC3 and LC4 for both imaging and behavior, as lower correlations across models were specifically observed for these components.”

“Although we could consider some socio-demographic variables and proxies of social inequalities relating to race and ethnicity as covariates in our main model, the relationship of these social factors to structural and functional brain phenotypes remains to be established with more targeted analyses.”

Data quality: for a relevant paper for ABCD, See: https://www.biorxiv.org/content/10.1101/2023.02.28.530498v1Including the Euler number (https://www.sciencedirect.com/science/article/pii/S1053811917310832) or the manual ratings from the ABCD preprint would mitigate these concerns. For dMRI data I would suggest including a summary measure of in-scanner motion as a covariate.

We controlled for T1w scan quality in all structural imaging metrics in a supplementary analysis and found results to be highly consistent with our main model. Similarly, additionally controlling for head motion during DWI scans yielded consistent findings to those presented in the manuscript.

We agree that data quality was not accounted for in our analysis of T1w- and diffusion-derived metrics. We now accounted for T1w image quality by adding manual quality control ratings to the regressors applied to all structural imaging metrics prior to performing the PLS analysis, and reported the consistency of this new model with original findings. See P.36, P.19:

“We also considered manual quality control ratings as a measure of T1w scan quality. This metric was included as a covariate in a multiple linear regression model accounting for potential confounds in the structural imaging data, in addition to age, age^2^, sex, site, ethnicity, ICV, and total surface area. Downstream PLS results were then benchmarked against those obtained from our main model.”

“Considering scan quality in T1w-derived metrics (from manual quality control ratings) yielded similar results to our main analysis, with an average correlation of 0.986±0.014 across imaging composite scores.”

As for diffusion imaging, we also regressed out effects of head motion in addition to age, age^2^, sex, site, and ethnicity from FA and MD measures and reported the consistency with our original results (P.36, P.19):

“We tested another model which additionally included head motion parameters as regressors in our analyses of FA and MD measures, and assessed the consistency of findings from both models.”

“Additionally considering head motion parameters from diffusion imaging metrics in our model yielded consistent results to those in our main analyses (mean r=0.891, S.D.=0.103; r=0.733-0.998).”

Reviewer #3 (Recommendations *for the authors):*

While reading the paper, I noticed that the authors have included a substantial amount of analyses. However, it was not entirely clear to me why certain data were utilized in each analysis. In order to enhance the clarity and comprehension of their work, I recommend the following improvements:

We thank the Reviewer for the helpful and thoughtful suggestions, which we addressed below.

- Provide a concise and explicit description of the data used for each analysis. This will help readers understand the specific datasets employed in each analysis and their relevance to each analytical approach.

We thank the Reviewer for this comment, and have now provided a general description of the analysis flow in the beginning of the *Results*, P.6*.*

“We divided a fully preprocessed and quality controlled subsample of the ABCD dataset that had structural and resting-state fMRI data available into Discovery (N=3,504, i.e., 2/3 of the dataset) and Replication (N=1,747 i.e., 1/3 of the dataset) subsamples, using randomized data partitioning with both subsamples being matched on age, sex, ethnicity, acquisition site, and overall psychopathology (i.e., scores of the first principal component derived from the 118 items of the CBCL). After applying dimensionality reduction to imaging features, we ran a PLS analysis in the Discovery subsample to associate imaging phenotypes and CBCL items. Significant components, identified using permutation testing, were comprehensively described, and we assessed associations to sensory-to-transmodal functional gradient organization for macroscale contextualization. We furthermore related findings to initially held out measures of white matter architecture and task-based fMRI patterns that were available in subsets of participants. Finally, we repeated our analyses in the Replication subsample and assessed generalizability when using Discovery-derived loadings in the Replication subsample.”

- Given that the structural and rsfMRI data were used for the main PLS analysis and the rest for validation, consider providing a more detailed explanation of the rationale behind this choice. Additionally, elucidate how these datasets contribute to the overall findings and conclusions of your study.

We thank the Reviewer for this comment, and further justify the rationale for our analytical approach. Please see P.6:

“Measures of brain structure and resting state fMRI were chosen for the main analyses as they (i) have been acquired in the majority of ABCD subjects, (ii) represent some of the most frequently acquired, and widely studied imaging phenotypes, and (iii) profile intrinsic gray matter network organization. Nevertheless, we also conducted post hoc analyses in smaller subsamples based on diffusion-based measures of fiber architecture (i.e., fractional anisotropy, mean diffusivity) and functional connectivity during tasks tapping into executive and reward processes.”

- Enhance validation procedures: if I'm correct, all the PLS inferences were done in sample, so I wonder about the transferability of their found results.

In accordance with a similar comment from Reviewers 1 and 2, we assess out-of-sample prediction by applying the PCA coefficients derived from the discovery cohort imaging data to the replication cohort imaging data. The resulting PCA scores and behavioral data were then z-scored using the mean and standard deviation of the replication cohort. The SVD weights derived from the discovery cohort were applied to the normalized replication cohort data to derive imaging and behavioral composite scores, which were used to recover the contribution of each imaging and behavioral variable to the LCs (*i.e.,* loadings). Out-of-sample replicability of imaging (mean r=0.681, S.D.=0.131) and behavioral (mean r=0.948, S.D.=0.022) loadings was generally high across LCs 1-5. This analysis is reported in the revised manuscript (P.18).

“Generalizability of reported findings was also assessed by directly applying PCA coefficients and latent components weights from the PLS analysis performed in the discovery cohort to the replication sample data. Out-of-sample prediction was overall high across LCs1-5 for both imaging (mean r=0.681, S.D.=0.131) and behavioral (mean r=0.948, S.D.=0.022) loadings.”

Furthermore, to strengthen their argument regarding the relationship between changes in the sensory-to-transmodal axis and behavioral factors, it would be advisable for the authors to directly correlate composite scores reflecting behavior with the gradients. Additionally, if these correlations turn out to be non-significant, it would be interesting for the authors to discuss the possible reasons behind these findings.

We agree with the Reviewer that investigating gradient-behavior relationships could offer additional insights into the cortical basis of psychiatric symptomatology. However, as discussed in point 3.5, the current analysis pipeline precludes this direct comparison which is performed on a region-by-region basis across the span of the cortical gradient. Indeed, the behavioral loadings are provided for each CBCL item, and not cortical regions.

Lastly, to address the interpretation concerns for the first latent component, I suggest the following:- Delve deeper into the observed loadings: Provide a detailed analysis of the specific loadings associated with LC1 and how they relate to impulse control problems over different train-test sets. By exploring the nuances of these loadings, the authors can offer a more precise understanding of the factor's nature and its connection to the broader p-factor construct.

We thank the Reviewer for this suggestion. We provide item-specific loadings for LC1-3 in Supplementary File 1b, and briefly explore these loadings in the Results section pertaining to *LC1 (P.8):*

“All symptom items loaded positively on LC1 – which is expected given prior data showing that every prevalent mental disorder loads positively on the p factor (Lahey et al., 2012). The top behavioral loadings include being inattentive/distracted, impulsive behavior, mood changes, rule breaking, and arguing (Figure 2b, see Supplementary File 1b for all behavior loadings).”- Acknowledge the ongoing discussion: Recognize the existing discourse within the field regarding the interpretation and utilization of the p-factor. Discuss the differing perspectives and highlight the points of contention, emphasizing the need for further investigation and clarification.

We take into account the Reviewer’s point by adding greater nuance to the *Discussion* concerning the use of the p factor on P.27.

“Other factors have also been suggested to impact the development of psychopathology, such as executive functioning deficits (Zelazo, 2020), earlier pubertal timing (Ullsperger and Nikolas, 2017), negative life events (Brieant et al., 2021), maternal depression (Goodman and Gotlib, 1999; Goodman et al., 2011), or psychological factors (e.g., low effortful control, high neuroticism, negative affectivity) (Lynch et al., 2021). Inclusion of such data could also help to add further insights into the rather synoptic proxy measure of the p factor itself (Fried et al., 2021), and to potentially assess shared and unique effects of the p factor vis-à-vis highly correlated measures of impulse control.”

[Editors’ note: what follows is the authors’ response to the second round of review.]

The manuscript has been improved but there are some remaining issues that need to be addressed, as outlined below:We thank the authors for their revised manuscript and their responses to the points raised by the reviewers. Before moving forward with further consideration of this revised submission, we would like to ask the authors to extend the out-of-sample analysis that has been provided and to report this more transparently because this was identified as a critical point by the reviewers in the last submission.

In accordance with the points raised below, we have expanded our reporting and discussion of the replication procedure in the revised manuscript.

More specifically, it seems that only the replicability of the in-sample and out-of-sample estimates is currently provided (and not the actual out-of-sample predictions) although this is not very clear from the description given (e.g. lines 388-390).

We have expanded the description of replication analyses in the revised manuscript (p.18-19). First, we provide results from an independent re-estimation of model parameters in the replication sample, and correlate imaging and behavioral loadings across samples (findings presented in Figure 7). As mentioned by the reviewers and the editors, we now explicitly acknowledge in the paper that this approach likely inflates estimated effect sizes due to model overfitting in each studied sample. We address this comment by assessing the consistency of imaging and behavioral loadings when applying model and sample statistics estimated from the discovery sample to the replication cohort. Taking into account point 3 of this response, we also now use sample statistics of the discovery sample for feature normalization (which was not the case in the previous response). We report associated statistics in the revised *Results* (p.18-19):

“We implemented different approaches to evaluate the robustness and potential generalizability of our findings. First, we performed a completely independent replication of the analysis pipeline in an unseen sample on participants (see Figure 7). We observed significant correlations between behavioral loadings of LCs 1-3 across discovery and replication samples (r=0.63-0.97). In terms of imaging loadings, RSFC loadings were replicated in LCs 1-3 (r=0.11-0.29, p_spin_<0.05); thickness loadings were replicated in LCs 1-3 (r=0.15-0.55, p_spin_<0.05) ; volume loadings were replicated in LCs 1-2 (r=0.18-0.19, p_spin_<0.05) but not LC3 (r=0.02, p=0.467); finally, surface area loadings were only replicated in LC2 (r=0.15, p_spin_=0.040). However, independently re-calculating model statistics in the replication sample may yield inflated effect sizes in estimating out-of-sample prediction. We address this limitation by applying all model weights computed in the discovery sample to the replication sample data. We first applied the imaging PCA coefficients computed in the discovery cohort to the replication cohort data. Resulting PCA scores and behavioral data were then normalized using the mean and standard deviation of corresponding data in the discovery cohort. Cross-validated composite scores were generated by multiplying singular value decompositions of the discovery cohort data with the normalized imaging PCA and behavioral data from the replication sample. Modality-specific and behavioral loadings were recovered by correlating cross-validated composite scores with normalized replication sample data. With this approach, we found that out-of-sample prediction was overall high across LCs1-3 for behavioral loading (r=0.94-0.97), and lower for imaging loadings (r=0.16-0.29). These analyses suggest that questionnaire item loadings were highly replicable across discovery and replication cohorts but indicate lower generalizability of structural and functional network loadings.”

Please report the out-of-sample prediction statistics (i.e. corresponding to the in-sample estimates currently shown in Figure 2A). Please also test the significance of these for example using permutation testing and adjust the Discussion section accordingly.

We now report cross-validated composite score correlations for LCs 1-3 (p.19):

“This lower replicability of brain features also affected out-of-sample prediction statistics linking imaging features and behavior (cross-validated composite scores), which were generally low across LCs but remained statistically significant (LC1 r=0.03; LC2 r=0.05; LC3 r=0.07; all permuted p<0.001 after permuting the first five LCs 10,000 times, accounting for site and FDR).”

Considering the low generalizability of model statistics to completely unseen data, we have also adjusted the discussion of these findings in the revised manuscript:

“Model generalizability to unseen data was limited by sample-specific variations in structural and functional imaging features, yet model parameters yielded statistically significant brain-behavior associations in unseen data.” (p.23)

“In this regard, our model was found to exhibit relatively poor out-of-sample prediction performance, as is often the case in explorations of complex brain-behavior relationships.” (p.27)

Additionally, please clarify the exact steps taken during the out-of-sample estimation procedure. As noted above, this is currently quite unclear. The standard approach within machine learning would be to keep the training and test sets completely independent, where any normalisation of the features prior to prediction is performed using statistics derived from the training set.

We significantly expanded the description of our approach as reported in our response to point 1.

[Editors’ note: what follows is the authors’ response to the third round of review.]

The manuscript has been improved but there are some remaining issues that need to be addressed, as outlined below:The authors have responded satisfactorily to most of the concerns raised by the reviewers. However there is one significant concern that must be addressed before we can consider this suitable for publication in eLife. The focus and narrative of the paper is still nearly entirely based around in-sample statistics, especially the canonical correlations reported in figures 2, 3 and 4 (r ~ 0.35). We feel that this is too optimistic and does not accurately reflect the true magnitude of the effects reported, even in view of the discovery-replication conducted. This is because the out-of-sample canonical correlations now (briefly) included in the manuscript are of a much smaller magnitude (r=0.03 – 0.07), indicating a very small amount of explained variance.We do not consider the low explained variance to be problematic per se, and indeed it is in line with current standards in the literature (e.g. https://www.nature.com/articles/s41586-022-04492-9), but it should be transparently and accurately reported. In general, given the well-established high propensity of CCA/PLS to overfit thus, resulting in quite brittle models especially where large number of predictor variables are included, we consider that in-sample canonical correlations are not appropriate indicators of model performance in neuroimaging and we should rely on out-of-sample statistics instead (see e.g. https://pubmed.ncbi.nlm.nih.gov/32224000/ for a discussion on this). It is perhaps also useful to note that this view is shared by the reviewers and the reviewing editor who assessed this manuscript.

We agree with the Editors that the use of out-of-sample statistics considerably reduced the effect sizes relative to within-sample model fitting, as expected. We acknowledge and agree with the Editors that methods such as PLS overfit to the analyzed data. We believe we have thoroughly explored and mitigated this limitation of the method through the numerous control analyses detailed in the paper that attest to the robustness of our main findings, despite reductions in observed effect sizes. These control analyses explore consistency of findings across different combinations of confound variables, replicate the model in an independent sample, and assess generalizability using out-of-sample statistics. As expected, this last set of analyses considerably reduced measured effect sizes. We believe to have adequately addressed the points raised by the Reviewers and the Editors in the two previous rounds of reviews which asked to perform and report this additional analysis.

To address this, please: (i) replace the in-sample statistics reported in all the relevant figures with out-of-sample statistics (or simply add the out of sample statistics to the figures), (ii) report the out-of-sample canonical correlations in the abstract, and (iii) adjust narrative of the paper (and where appropriate downstream analyses) accordingly to focus principally on the out-of-sample statistics.

In the interest of transparency, we now clearly report out-of-sample statistics in the *Abstract*, in Figures 2-4, their corresponding captions, as well as the revised *Results*, alongside the within-sample statistics (addressing points i and ii). As the data in the figures is generated from within-sample analyses, we believe there is value in reporting these within-sample effects.

Regarding point iii, we now report out-of-sample statistics alongside within-sample statistics in the Results. We also discuss this effect size drop in the first paragraph of the Discussion (p.23):

“Model generalizability to unseen data was overall low and was likely limited by sample-specific variations in structural and functional imaging features. Although model parameters yielded statistically significant brain-behavior associations in unseen data over LCs 1-3, the poor generalizability of model parameters strongly mitigates the potential of presented neuroimaging signatures to serve screening or diagnostic purposes in detecting childhood psychopathology. Symptom dimensions were consistent with prior literature, but independent replication of the model indicated strong sample-specific variations in structural and functional imaging features which may explain poor generalizability.”

Repeating all downstream analyses and regenerating associated figures from the out-of-sample data at this point in the review stage would significantly hinder the timeliness of our submission of a revised manuscript. We hope the Editors will be understanding in this regard and accept a middle-ground on this suggestion in which we nuanced the reporting of findings throughout the revised paper considering the lower effect sizes.